# DenseDPO: Fine-Grained Temporal Preference Optimization for Video Diffusion Models

**Ziyi Wu**[1,2,3], **Anil Kag**[1], **Ivan Skorokhodov**[1], **Willi Menapace**[1], **Ashkan Mirzaei**[1],
**Igor Gilitschenski**[2,3,*], **Sergey Tulyakov**[1,*], **Aliaksandr Siarohin**[1,*]
[1]Snap Research, [2]University of Toronto, [3]Vector Institute

## Abstract

Direct Preference Optimization (DPO) has recently been applied as a post-training technique for text-to-video diffusion models. To obtain training data, annotators are asked to provide preferences between two videos generated from independent noise. However, this approach prohibits fine-grained comparisons, and we point out that it biases the annotators towards low-motion clips as they often contain fewer visual artifacts. In this work, we introduce DenseDPO, a method that addresses these shortcomings by making three contributions. First, we create each video pair for DPO by denoising corrupted copies of a ground truth video. This results in aligned pairs with similar motion structures while differing in local details, effectively neutralizing the motion bias. Second, we leverage the resulting temporal alignment to label preferences on short segments rather than entire clips, yielding a denser and more precise learning signal. With only one-third of the labeled data, DenseDPO greatly improves motion generation over vanilla DPO, while matching it in text alignment, visual quality, and temporal consistency. Finally, we show that DenseDPO unlocks automatic preference annotation using off-the-shelf Vision Language Models (VLMs): GPT accurately predicts segment-level preferences similar to task-specifically fine-tuned video reward models, and DenseDPO trained on these labels achieves performance close to using human labels. Additional results are available at https://snap-research.github.io/DenseDPO/.

## 1 Introduction

Recent advances in diffusion models [23] have enabled high-quality text-guided video generation [2, 7, 24, 35, 58, 66, 71]. Despite tremendous progress, existing video generators still fall short on temporal coherence, visual fidelity, and prompt alignment [89], impeding their industry-level applications.

Inspired by the success of learning from human feedback in language models [3, 52] and image diffusion [5, 17, 70], recent works have explored preference alignment in video diffusion [40, 60, 87]. Among them, methods based on Direct Preference Optimization (DPO) [61] stand out as they bypass the need for an explicit reward model [10, 47, 50, 65]. However, existing DPO methods for video diffusion are largely adapted from their image-based counterparts, without addressing the unique challenges inherent to video generation. Typically, these methods first generate videos from *independent* noise maps, followed by human preference labeling to construct comparison pairs. Yet, human preferences in video are influenced by multiple, sometimes inversely correlated, factors, such as the visual quality (i.e., pixel-level fidelity) and the dynamic degree (i.e., strength of global motion). Indeed, current video generation models excel at producing high-quality slow-motion videos, while struggling to synthesize more challenging dynamic scenes [8]. As a result, when annotators are asked to express preferences, they often favor artifact-free slow-motion clips. Applying DPO training on such preference data further reinforces video generators' bias toward slow-motion content, ultimately suppressing the model's ability to generate dynamic and motion-rich videos.

39th Conference on Neural Information Processing Systems (NeurIPS 2025).

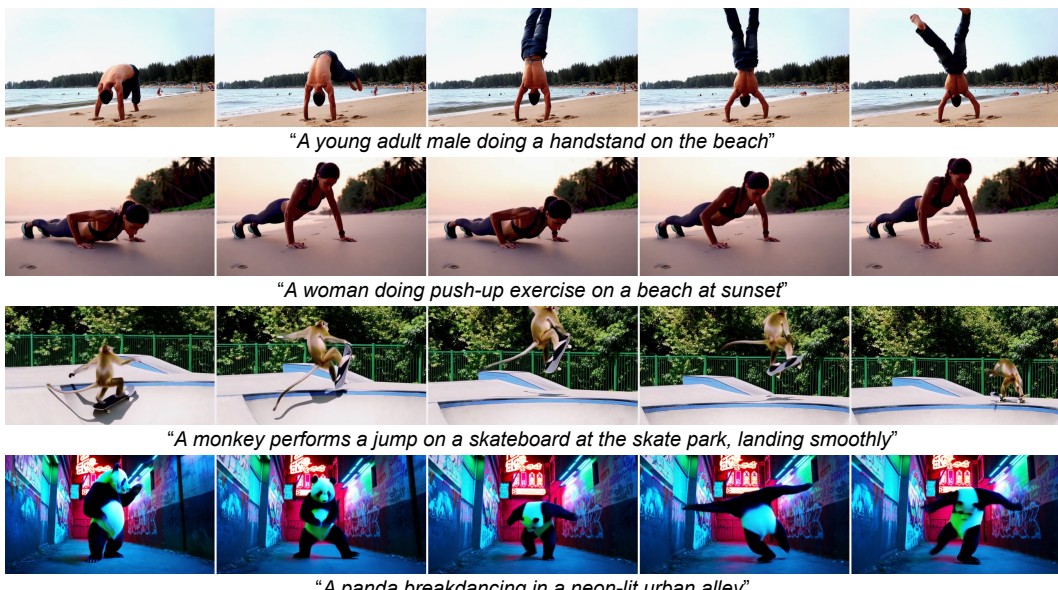

*"A young adult male doing a handstand on the beach"*

*"A woman doing push-up exercise on a beach at sunset"*

*"A monkey performs a jump on a skateboard at the skate park, landing smoothly"*

*"A panda breakdancing in a neon-lit urban alley"*

Figure 1: **Text-to-video results with our DenseDPO aligned model.** Our method improves both visual quality and temporal consistency of the model, enabling generation of challenging motion.

A natural approach to enhance competing factors, drawing inspiration from Pareto optimization, is to fix some attributes within each video pair while varying others. Motivated by the guided image synthesis approach in SDEdit [51], we generate a pair by introducing different partial noise to a ground-truth video and perform denoising. The resulting videos in each pair share high-level semantics and motion trajectories while differing in local visual details [8, 72], allowing us to reduce spurious correlations. However, guided sampling inherently reduces the diversity across generated video pairs, leading to degraded DPO performance [53, 63]. A straightforward solution might be to annotate more data pairs. Instead, we propose extracting richer and more accurate supervision from each video pair by collecting segment-level preference labels.

Prior works show that multi-dimensional scores are superior to a single label in preference alignment [42, 80, 93]. Unlike images, videos have a unique temporal dimension [78]. In practice, we observe that human preferences over video pairs often vary across time, as artifacts may appear at different timestamps in each video, leading to inconsistent preferences. This issue is more severe with modern video generators as they produce longer videos. Therefore, we split videos into short segments (e.g., 1s slices), and collect per-segment preference labels. Thanks to temporally aligned videos from guided sampling, there is a clear one-to-one correspondence between segment pairs, simplifying the annotation process. Segment feedback also reduces the amount of ties when both videos contain artifacts, and provides more accurate supervision. In addition, it allows us to apply existing vision-language models (VLMs) [1, 4] which can produce reliable judgment on short segments.

**Our main contributions** are threefold: **(i)** A DenseDPO framework tailored towards video generation, with improved data construction and preference granularity over vanilla DPO; **(ii)** DenseDPO retains the motion strength of the base model while matching other metrics of vanilla DPO, with significantly higher data efficiency; **(iii)** We show that existing VLMs fail to label preference over long videos (e.g., 5s), but they perform well in segment-level preference, achieving results close to human labels.

## 2   Related Work

**Preference learning for image diffusion.** Inspired by the success of human feedback learning in language modeling [3, 52], similar approaches have been adapted to image generation. One major line of work focuses on training reward models from human preference labels [33, 75, 76, 79, 93], these models can be used as loss functions to optimize generators by direct gradient backpropagation [9, 13, 31, 59, 77, 79] or policy gradients [5, 17]. Another line of work utilizes predicted rewards on training data to re-weight the diffusion loss [38], or trains only on high-scoring samples [14, 15, 39]. However, all these methods require an explicitly trained reward model, and may suffer from the reward hacking issue [13, 79]. In contrast, Diffusion-DPO [70] and D3PO [83] directly optimize the model on

pre-collected human preference pairs, bypassing the need for online reward feedback. Building on this paradigm, subsequent works have explored improving comparison data pairs [28, 32, 88, 90], better DPO objectives [29, 41, 73], and credit assignment over denoising timesteps [43, 94].

**Preference learning for video diffusion.** Early approaches to preference learning in video diffusion directly borrow techniques from image diffusion, such as direct reward optimization [40, 56, 60, 68, 87] and training loss re-weighting [19]. However, they often rely on image reward models [33, 75] to provide supervision. Recent papers thus focus on developing better video reward models [46, 47, 80]. One strategy aggregates multiple video quality assessment metrics [21, 30] to a final score [47, 91]. However, existing metrics are only effective for short videos [46, 80], limiting their applicability for modern video generators that produce long videos [7, 71, 84]. To address this limitation, LiFT [74], VisionReward [80], and VideoAlign [46] collect a large number of videos from advanced video generators, label human feedback, and fine-tune VLMs to predict preferences. With a powerful video reward model, they apply weighted training [74] or DPO [46, 80] to improve video generation. In contrast to prior works, we focus on DPO for video diffusion using *direct* human annotations, i.e., without an explicit reward model. Analogous to the verbosity bias observed in language model preference learning, where annotators favor longer outputs [64, 67], we identify a motion bias in video preference labels, where slow-motion videos are often preferred. To mitigate this, we propose a better data pair construction strategy to address this bias via guided video generation.

**Rich feedback for alignment.** While early preference alignment methods treated human feedback as a single binary label, recent works begin to exploit rich, multi-dimensional feedback [42, 80, 92]. In image generation, MPS [93] learns a reward model that evaluates images on four dimensions including aesthetics, semantics, detail, and overall quality, improving its alignment with humans. On the other hand, Liang et al. [42] curates a dataset that localizes regions of artifacts and misaligned words in the text prompt, leading to better DPO performance. Multi-aspect feedback is even more critical for video generation due to its inherently higher dimensionality. Recent works all explicitly model dimensions such as visual fidelity, text relevance, and motion consistency [46, 47, 80]. However, a notable limitation is that they still aggregate feedback at the whole-video level, neglecting the fine-grained temporal dimension of preferences. In contrast, our DenseDPO partitions videos into short, temporally aligned segments, and collects preferences for each segment. This is conceptually similar to the sentence-level preference label used in language models [34, 86]. By localizing feedback to brief windows, we obtain more accurate and denser supervision signals for DPO training.

## 3 Method

We build upon diffusion models and the standard Direct Preference Optimization (DPO) framework, which we refer to as VanillaDPO (Sec. 3.1). We discuss the motion bias inherent in using this naïve approach for video generation and introduce StructuralDPO, a method that optimizes human preferences on structurally similar video pairs (Sec. 3.2). To address the reduction of diversity induced from using structurally similar videos, we propose DenseDPO, which enables fine-grained human preference alignment along the temporal axis of videos (Sec. 3.3).

### 3.1 Background: Video Diffusion and DPO

**Rectified-flow diffusion models.** Let $\boldsymbol{x} \in \mathbb{R}^{T \times H \times W}$ denote a video sample of length $T$ with spatial dimensions $H \times W$. We follow the rectified flow framework [44, 48], which learns a transport map from the standard normal distribution $\boldsymbol{\epsilon} \sim \mathcal{N}(\mathbf{0}, \boldsymbol{I})$ to the distribution of real videos $\boldsymbol{x} \sim p_{\text{data}}$ with a denoiser. The forward diffusion process produces a noisy input $\boldsymbol{x}_t$ at time $t \in [0, 1]$ via a linear interpolation with noise $\boldsymbol{\epsilon}$: $\boldsymbol{x}_t = (1 - t)\boldsymbol{x}_0 + t\boldsymbol{\epsilon}$. The denoiser $\mathbf{G}_{\boldsymbol{\theta}}(\boldsymbol{x}_t, t, \boldsymbol{c})$, implemented as a neural network parameterized by $\boldsymbol{\theta}$, is trained to reverse this process with the following objective:

$$\min_{\boldsymbol{\theta}} \mathbb{E}_{t \sim p(t), \boldsymbol{x} \sim p_{\text{data}}, \boldsymbol{\epsilon} \sim \mathcal{N}(\mathbf{0}, \boldsymbol{I})} \|(\boldsymbol{\epsilon} - \boldsymbol{x}) - \mathbf{G}_{\boldsymbol{\theta}}(\boldsymbol{x}_t, t, \boldsymbol{c})\|^2, \tag{1}$$

where $p(t)$ is the distribution of noise levels (following [16], we adopt the logit-normal one) and $\boldsymbol{c}$ refers to the auxiliary conditioning variable such as text embeddings.

**VanillaDPO.** In the direct preference optimization framework [61, 70], a generative model is trained to align its outputs with human preferences. Typically, these preferences are defined by a dataset $\mathcal{D} = \{(\boldsymbol{c}, \boldsymbol{x}^0, \boldsymbol{x}^1, l)\}$, where each sample consists of two videos $\{\boldsymbol{x}^0, \boldsymbol{x}^1\}$ per input condition $\boldsymbol{c}$ and

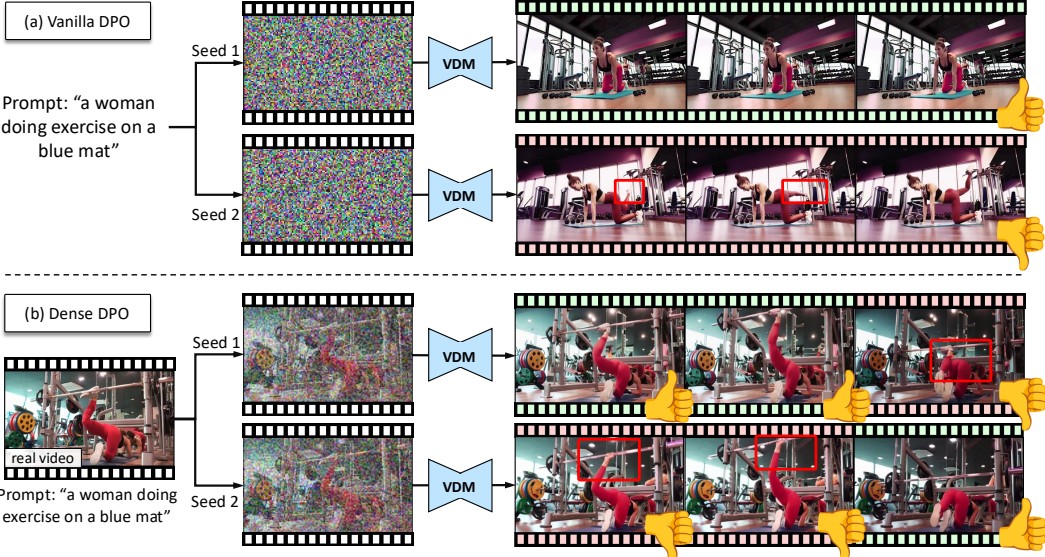

Figure 2: **Comparison between VanillaDPO (top) and DenseDPO (bottom).** VanillaDPO compares two videos generated from independent random noises and only assigns a single binary preference, biasing the annotators toward slow-motion videos. In contrast, DenseDPO generates structurally similar videos from partially noised real videos, and label segment-level dense preferences.

their preference label $l \in \{-1, +1\}$. The preference function is defined as:

$$l(\boldsymbol{x}^0, \boldsymbol{x}^1) = \begin{cases} +1, & \text{if } \boldsymbol{x}^0 \succ \boldsymbol{x}^1 \quad (\text{i.e., } \boldsymbol{x}^0 \text{ is preferred over } \boldsymbol{x}^1) \\ -1, & \text{if } \boldsymbol{x}^1 \succ \boldsymbol{x}^0 \quad (\text{i.e., } \boldsymbol{x}^1 \text{ is preferred over } \boldsymbol{x}^0) \end{cases} \quad (2)$$

The Bradley-Terry (BT) model [6] defines pairwise preference using a reward function $r(\boldsymbol{x}, \boldsymbol{c})$ which computes the alignment score between the sample $\boldsymbol{x}$ and the input condition $\boldsymbol{c}$. The corresponding probabilistic preference can be expressed as:

$$p_{\text{BT}}(\boldsymbol{x}^0 \succ \boldsymbol{x}^1 | \boldsymbol{c}) = \sigma(r(\boldsymbol{x}^0, \boldsymbol{c}) - r(\boldsymbol{x}^1, \boldsymbol{c})); \quad p_{\text{BT}}(\boldsymbol{x}^1 \succ \boldsymbol{x}^0 | \boldsymbol{c}) = \sigma(r(\boldsymbol{x}^1, \boldsymbol{c}) - r(\boldsymbol{x}^0, \boldsymbol{c})), \quad (3)$$

where $\sigma(\cdot)$ is the sigmoid function.

Rafailov et al. [61] defines the binary preference optimization as explicitly optimizing the binary reward objective $\log \sigma(l(\boldsymbol{x}^0, \boldsymbol{x}^1) * (r(\boldsymbol{x}^0, \boldsymbol{c}) - r(\boldsymbol{x}^1, \boldsymbol{c})))$ in conjunction with a Kullback-Leibler (KL) divergence regularization to control the deviation from a reference model. Wallace et al. [70] re-formulated the preference optimization framework for diffusion models assuming the presence of a reference model $\mathbf{G}_{\text{ref}}$, which is further extended to rectified flow models in [46]. Given a sample $(\boldsymbol{c}, \boldsymbol{x}^0, \boldsymbol{x}^1, l)$, the denoiser $\mathbf{G}_{\boldsymbol{\theta}}$, and the reference model $\mathbf{G}_{\text{ref}}$, we can define an *implicit* reward as:

$$\boldsymbol{s}(\boldsymbol{x}^*, \boldsymbol{c}, t, \boldsymbol{\theta}) = \|(\boldsymbol{\epsilon}^* - \boldsymbol{x}^*) - \mathbf{G}_{\boldsymbol{\theta}}(\boldsymbol{x}_t^*, t, \boldsymbol{c})\|_2^2 - \|(\boldsymbol{\epsilon}^* - \boldsymbol{x}^*) - \mathbf{G}_{\text{ref}}(\boldsymbol{x}_t^*, t, \boldsymbol{c})\|_2^2, \quad (4)$$

where $\boldsymbol{x}_t^* = (1 - t)\boldsymbol{x}^* + t\boldsymbol{\epsilon}^*, \boldsymbol{\epsilon}^* \sim \mathcal{N}(\boldsymbol{0}, \boldsymbol{I})$ is a noisy latent for input $\boldsymbol{x}^*$ (either $\boldsymbol{x}^0$ or $\boldsymbol{x}^1$) at time $t$. With the implicit reward function, the VanillaDPO objective is defined as follows:

$$\mathcal{L}(\boldsymbol{\theta}) = -\mathbb{E}_{\substack{(\boldsymbol{c}, \boldsymbol{x}^0, \boldsymbol{x}^1, l) \sim \mathcal{D}, \\ t \sim p(t), \boldsymbol{\epsilon} \sim \mathcal{N}(\boldsymbol{0}, \boldsymbol{I})}} \left[ \log \sigma \left( -\beta * l(\boldsymbol{x}^0, \boldsymbol{x}^1) * \left( \boldsymbol{s}(\boldsymbol{x}^0, \boldsymbol{c}, t, \boldsymbol{\theta}) - \boldsymbol{s}(\boldsymbol{x}^1, \boldsymbol{c}, t, \boldsymbol{\theta}) \right) \right) \right]. \quad (5)$$

### 3.2 StructuralDPO: Preference Learning over Structurally Similar Videos

**Motion bias in VanillaDPO.** In the standard VanillaDPO pipeline, preference pairs are created by independently sampling two videos $(\boldsymbol{x}^0, \boldsymbol{x}^1)$ from different noise seeds under the same conditioning $\boldsymbol{c}$ (see Algo. 1), followed by human preference annotation. While this approach works reasonably well for images, its direct extension to videos introduces new issues due to the presence of the new temporal dimension. Independent noises often result in videos with significantly different motion patterns and global layouts (see Fig. 2 (a)). For example, in a typical preference pair, one video may be nearly static but visually clean, while the other contains the desired motion but also introduces artifacts, such as distorted limbs or flickering.

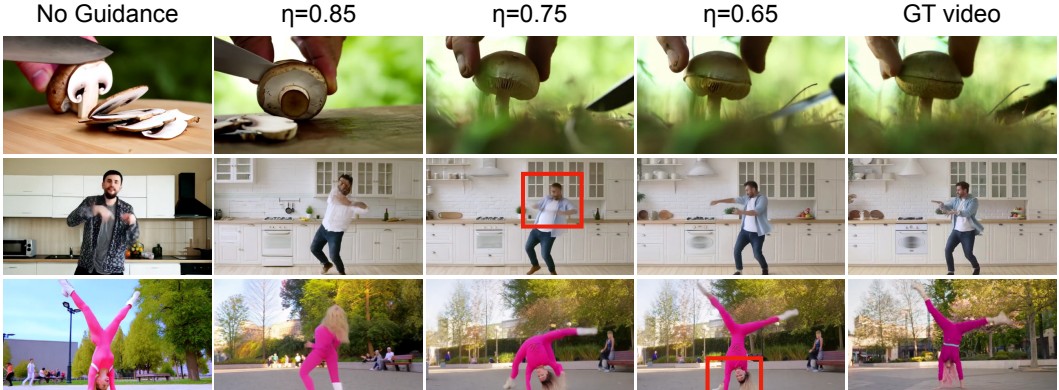

| No Guidance | η=0.85 | η=0.75 | η=0.65 | GT video |

Figure 3: **Guided video generation with different $\eta$.** Lower $\eta$ means more guidance. We sample one frame per video for visualization. $\eta = 0.75$ is enough to maintain the motion trajectory and high-level semantics of the ground-truth video. For slow-motion videos (top), a high $\eta$ suffices to generate artifact-free videos, while videos with challenging motion (bottom) require more guidance.

We empirically observe that this is a common bias in generated video data. Indeed, video models often excel at producing high-quality slow-motion clips, while dynamic videos usually contain visible artifacts [8, 89]. Since humans typically perceive clean static videos as more realistic than artifact-prone dynamic ones, this often leads to a preference dataset that systematically over-represents static content. Consequently, a model trained with DPO on this dataset would produce videos with reduced motion. In our preliminary DPO experiments, we observed a substantial drop in dynamic degree using our base model (see Tab. 1 and Tab. 2). The same issue has also been observed in prior works, e.g., Tab.10 of [80] using CogVideoX [84] and Tab. 2 of [47] using VideoCrafter-v2 [11].

**StructuralDPO.** To address the static bias in VanillaDPO, we propose StructuralDPO, which modifies the data curation strategy using guided generation [51] to obtain pairs of videos with similar motion trajectories. Specifically, in VanillaDPO, we start from independent noise, i.e., $\boldsymbol{x}_N^0 \sim \mathcal{N}(\mathbf{0}, \boldsymbol{I})$; $\boldsymbol{x}_N^1 \sim \mathcal{N}(\mathbf{0}, \boldsymbol{I})$ and then denoise from step $N$ till step 1. Instead, we propose to denoise from a partially noised real video to generate video pairs. Concretely, given a ground truth video $\boldsymbol{x}$ and the guidance level $\eta \in [0, 1]$, we obtain the corrupted noisy video at step $n = \text{round}(\eta * N)$ as:

$$\boldsymbol{x}_n^0 = (1 - \eta)\boldsymbol{x} + \eta\boldsymbol{\epsilon}^0; \quad \boldsymbol{x}_n^1 = (1 - \eta)\boldsymbol{x} + \eta\boldsymbol{\epsilon}^1; \quad \text{where } \boldsymbol{\epsilon}^0, \boldsymbol{\epsilon}^1 \sim \mathcal{N}(\mathbf{0}, \boldsymbol{I}). \tag{6}$$

The corrupted videos are then denoised from step $n$ to 1, as outlined in Algo. 2. Here, $\eta$ governs the structural similarity between the two samples. Since early diffusion steps control the global motion [8], this approach preserves the overall dynamics while allowing variations in local details. We compare videos of varying motion strengths generated with different $\eta$ in Fig. 3.

StructuralDPO applies the standard DPO formulation (Eq. (5)) on this structurally consistent dataset, which helps to focus the preference on the temporal artifacts and visual inconsistencies, anchoring other dimensions like dynamic degree. Additionally, guided sampling simplifies the generation task, allowing the model to produce artifact-free highly dynamic videos more reliably. Finally, guided sampling reduces data construction costs as it requires fewer sampling steps.

---

**Algorithm 1** Vanilla Paired Video Generation

**Input:** Denoiser $\mathbf{G}_{\boldsymbol{\theta}}$, Input Condition $\boldsymbol{c}$, Inference Steps $N$
**Init:** $\Delta_t = \frac{1}{N}$
**Init:** $\boldsymbol{x}_N^0 \sim \mathcal{N}(\mathbf{0}, \boldsymbol{I})$
**Init:** $\boldsymbol{x}_N^1 \sim \mathcal{N}(\mathbf{0}, \boldsymbol{I})$
**for** $i = N$ **to** 1 **do**
  $t = \frac{i}{N}$
  $\boldsymbol{x}_{i-1}^0 = \boldsymbol{x}_i^0 + \mathbf{G}_{\boldsymbol{\theta}}(\boldsymbol{x}_i^0, t, \boldsymbol{c}) * \Delta_t$
  $\boldsymbol{x}_{i-1}^1 = \boldsymbol{x}_i^1 + \mathbf{G}_{\boldsymbol{\theta}}(\boldsymbol{x}_i^1, t, \boldsymbol{c}) * \Delta_t$
**end for**
**return** Video Pair $(\boldsymbol{x}_0^0, \boldsymbol{x}_0^1)$

**Algorithm 2** Guided Paired Video Generation

**Input:** Denoiser $\mathbf{G}_{\boldsymbol{\theta}}$, Input Condition $\boldsymbol{c}$, Inference Steps $N$, Real Video $\boldsymbol{x}$, Guidance Level $\eta \in [0, 1]$
**Init:** $\Delta_t = \frac{1}{N}$, $n = \text{round}(\eta * N)$
**Init:** $\boldsymbol{x}_n^0 = (1 - \eta)\boldsymbol{x} + \eta\boldsymbol{\epsilon}^0, \boldsymbol{\epsilon}^0 \sim \mathcal{N}(\mathbf{0}, \boldsymbol{I})$
**Init:** $\boldsymbol{x}_n^1 = (1 - \eta)\boldsymbol{x} + \eta\boldsymbol{\epsilon}^1, \boldsymbol{\epsilon}^1 \sim \mathcal{N}(\mathbf{0}, \boldsymbol{I})$
**for** $i = n$ **to** 1 **do**
  $t = \frac{i}{N}$
  $\boldsymbol{x}_{i-1}^0 = \boldsymbol{x}_i^0 + \mathbf{G}_{\boldsymbol{\theta}}(\boldsymbol{x}_i^0, t, \boldsymbol{c}) * \Delta_t$
  $\boldsymbol{x}_{i-1}^1 = \boldsymbol{x}_i^1 + \mathbf{G}_{\boldsymbol{\theta}}(\boldsymbol{x}_i^1, t, \boldsymbol{c}) * \Delta_t$
**end for**
**return** Video Pair $(\boldsymbol{x}_0^0, \boldsymbol{x}_0^1)$

### 3.3 DenseDPO: Rich Temporal Feedback with Segment-Level Preferences

Although StructuralDPO effectively preserves dynamic degree, models trained with it tend to generate videos with lower visual quality and weaker text alignment compared to VanillaDPO. This performance gap is because video pairs are structurally similar, which reduces diversity in the curated DPO dataset, and, as we discuss in Appendix B, can unintentionally drive the model to diverge from the real data distribution [53, 63]. A straightforward solution is to obtain more labeled data, but it increases the annotation cost compared to VanillaDPO. Instead, we explore an alternative approach to increase data and annotation diversity without increasing the number of labeled video pairs.

**DenseDPO.** In VanillaDPO and StructuralDPO, a scalar preference label $l \in \{-1, +1\}$ is obtained for the entire video of length $T$. Instead, DenseDPO annotates preferences on shorter temporal segments. Since guided video generation (Algo. 2) yields structurally similar video pairs, the same time period in both videos has a clear correspondence, making comparison feasible. We show an example in Fig. 2 (b) with the intervals being a single frame: for frame 1 and 2, the first video is better, while for frame 3, the second video is better. Formally, given two videos $(\boldsymbol{x}^0, \boldsymbol{x}^1)$ and the interval length $s$, we can break down videos into $F = \mathrm{ceil}(\frac{T}{s})$ temporal segments of length $s$ by splitting along the time dimension. The resulting video pairs $(\{\boldsymbol{x}_f^0, \boldsymbol{x}_f^1\}_{f=1}^F)$ are annotated with preferences over each segment, yielding segment-level dense preference labels $\boldsymbol{l} \in \{-1, +1\}^F$, i.e., $\boldsymbol{l}(\boldsymbol{x}^0, \boldsymbol{x}^1) = [l(\boldsymbol{x}_f^0, \boldsymbol{x}_f^1)]_{f=1}^F$. Thus, following Eq. (5), we can formulate the DenseDPO objective as:

$$\mathcal{L}(\boldsymbol{\theta}) = -\mathbb{E}_{\substack{(\boldsymbol{c}, \boldsymbol{x}^0, \boldsymbol{x}^1, \boldsymbol{l}) \sim \mathcal{D} \\ t \sim p(t), \, \boldsymbol{\epsilon} \sim \mathcal{N}(\boldsymbol{0}, \boldsymbol{I})}} \log \sigma \left( -\beta \sum_{f=1}^F l(\boldsymbol{x}_f^0, \boldsymbol{x}_f^1) * \left( \boldsymbol{s}(\boldsymbol{x}^0, \boldsymbol{c}, t, \boldsymbol{\theta})_f - \boldsymbol{s}(\boldsymbol{x}^1, \boldsymbol{c}, t, \boldsymbol{\theta})_f \right) \right), \quad (7)$$

where $\boldsymbol{s}(\cdot)_f$ is the implicit reward value on the $f$-th video segment.

In our collected dense preference data, we find that over 60% of video pairs have both winning and losing labels in $\boldsymbol{l}$. In regular preference annotation such pairs will either be treated as ties or choose the video with fewer artifacts. In the latter case, this encourages the model to minimize loss on videos with artifacts in Eq. (5), degrading the model performance. In contrast, DenseDPO assigns preference labels more accurately over time, only optimizing models on segments with a clear difference.

**Segment preference annotation with VLMs.** Another benefit of the DenseDPO is that it allows us to use off-the-shelf VLMs for automatic preference labeling. Prior works point out that existing VLMs struggle at assessing long videos (e.g., 5s) [21, 80], often requiring task-specific fine-tuning and large-scale human annotations to train effective video reward models. Instead, we show that pre-trained VLMs are already capable of processing short clips (e.g., 1s). Given two temporally aligned videos, we feed in pairs of segments into a VLM, and ask it to identify the better one. As we will demonstrate in the experiments (Tab. 3), GPT-o3 [1] achieves high accuracy on segment-level preferences, leading to DPO results competitive with using human preference labels.

## 4 Experiments

Our experiments aim to answer the following questions: **(i)** How does DenseDPO perform against VanillaDPO? (Sec. 4.2) **(ii)** Can we leverage existing VLMs to produce high-quality preference labels? (Sec. 4.3) **(iii)** What is the impact of each component in our framework? (Sec. 4.4)

### 4.1 Experimental Setup

We list some key aspects of our experimental setup here. For full details, please refer to Appendix A.

**Preference learning data.** We curate a high-quality video dataset from existing datasets, resulting in around 55k videos. This is done by filtering the length, visual quality, and motion score of videos similar to [58], and prompting GPT-4o [3] to classify if the text prompt contains events of meaningful dynamics. This naturally gives us text prompts and corresponding ground-truth videos.

**Baselines.** Our pre-trained text-to-video generator is a DiT [55]-based latent flow model. We fine-tune it on the curated high-quality data, termed the *SFT* baseline. For *VanillaDPO*, we randomly select 30k text prompts from the curated dataset, generate 2 videos of 5s per prompt with Algo. 1, and ask human labelers to annotate preferences. This leads to around 10k winning-losing pairs after

Table 1: **Quantitative results on VideoJAM-bench [8].** We report automatic metrics from VBench [30] and VisionReward [80]. DenseDPO significantly outperforms Vanilla DPO in dynamic degree, while achieves similar performance in other dimensions.

| Method | VBench Metrics | | | | | | VisionReward Metrics | | | |
|---|---|---|---|---|---|---|---|---|---|---|
| | Aesthetic Quality | Imaging Quality | Subject Consistency | Background Consistency | Motion Smoothness | Dynamic Degree | Text Alignment | Visual Quality | Temporal Consistency | Dynamic Degree |
| Pre-trained | 54.65 | 55.85 | 88.29 | 91.50 | 92.40 | 84.16 | 0.770 | 0.192 | 0.354 | **0.680** |
| SFT | 55.19 | 53.26 | 87.71 | 91.52 | 92.72 | 83.25 | 0.773 | 0.205 | 0.279 | 0.675 |
| Vanilla DPO [46] | **57.25** | 60.38 | 91.21 | **93.94** | 93.43 | 80.25 | **0.867** | 0.371 | **0.636** | 0.535 |
| Structural DPO | 56.38 | 59.78 | 90.21 | 92.34 | 92.94 | 84.69 | 0.843 | 0.341 | 0.602 | 0.652 |
| **DenseDPO** | 56.99 | **60.92** | **91.54** | 93.84 | **93.56** | **85.38** | 0.863 | **0.376** | 0.632 | 0.680 |

Table 2: **Quantitative results on MotionBench.** We report automatic metrics from VBench [30] and VisionReward [80]. DenseDPO achieves similar motion smoothness compared to Vanilla DPO, while consistently outperforms it in visual quality, dynamic degree, and text alignment.

| Method | VBench Metrics | | | | | | VisionReward Metrics | | | |
|---|---|---|---|---|---|---|---|---|---|---|
| | Aesthetic Quality | Imaging Quality | Subject Consistency | Background Consistency | Motion Smoothness | Dynamic Degree | Text Alignment | Visual Quality | Temporal Consistency | Dynamic Degree |
| Pre-trained | 56.21 | 56.26 | 88.23 | 91.67 | 93.56 | 83.69 | 0.261 | 0.112 | 0.154 | 0.840 |
| SFT | 56.16 | 55.54 | 87.94 | 92.42 | 94.44 | **84.93** | 0.273 | 0.105 | 0.129 | 0.845 |
| Vanilla DPO [46] | 57.51 | 61.20 | 91.45 | 93.49 | **97.43** | 72.55 | 0.355 | 0.172 | **0.239** | 0.709 |
| Structural DPO | 57.46 | 59.84 | 90.98 | 93.13 | 97.11 | 79.95 | 0.347 | 0.152 | 0.229 | 0.839 |
| **DenseDPO** | **57.54** | **61.52** | **91.60** | **93.72** | 97.33 | 84.73 | **0.359** | **0.179** | 0.232 | **0.858** |

removing ties. For *StructuralDPO*, we use the same 30k prompts from VanillaDPO, and label human preferences on videos generated using Algo. 2. The guidance level $\eta$ is randomly sampled from $[0.65, 0.8]$ to obtain video pairs with similar motion. This again leads to around 10k winning-losing pairs. Both VanillaDPO and StructuralDPO apply the Flow-DPO loss in Eq. (5). Following prior works [46, 80], we set $\beta$ to 500 and apply LoRA [27] with rank 128 to fine-tune the video model. We train with the AdamW optimizer [49] and a global batch size of 256 for 1000 steps.

**DenseDPO implementation details.** For fair comparison with baselines, we only take 10k video pairs from the StructuralDPO training data to label dense preferences, which costs a similar amount of human annotation time. The segment length $s$ is set to 1s. Overall, more than 80% of video pairs have at least 1 non-tie segment and can be used in DPO training, greatly improving the data efficiency over using global preferences. All other hyper-parameters are the same as DPO baselines.

**Evaluation datasets.** We utilize two benchmarks to evaluate the performance of text-to-video generation. *VideoJAM-bench* [8] contains 128 prompts focusing on real-world scenarios with challenging motion, ranging from human actions to physical phenomena. We also construct *MotionBench*, which collects more diverse prompts from existing prompt sets [35, 58, 74] such as MovieGenBench. We run GPT-4o to select prompts with dynamic human actions, resulting in 419 prompts.

**Evaluation metrics.** We aim to measure the visual quality, text alignment, and motion quality of videos. Specifically, we want to evaluate both the smoothness and strength of the motion. Therefore, we adopt *VBench* [30] and a state-of-the-art video quality assessment model, *VisionReward* [80].

## 4.2 DPO with Human Labels

Tab. 1 and Tab. 2 present the quantitative results on VideoJAM-bench and MotionBench. Fig. 4 shows a qualitative comparison. VanillaDPO greatly improves the pre-trained model and the SFT baseline in all dimensions except dynamic degree due to the motion bias in video preference data. StructuralDPO retains the motion with paired video generation, while compromising the visual quality and text alignment. With the rich temporal feedback, DenseDPO consistently outperforms StructuralDPO. In addition, it matches all aspects of VanillaDPO and scores a significantly higher dynamic degree, despite using only one-third of labeled videos (10k *vs.* 30k).

Please refer to Appendix C.1 for comparison with more baselines including online RL-based methods. For more qualitative results, please check out Appendix C.2 and our project page for video results.

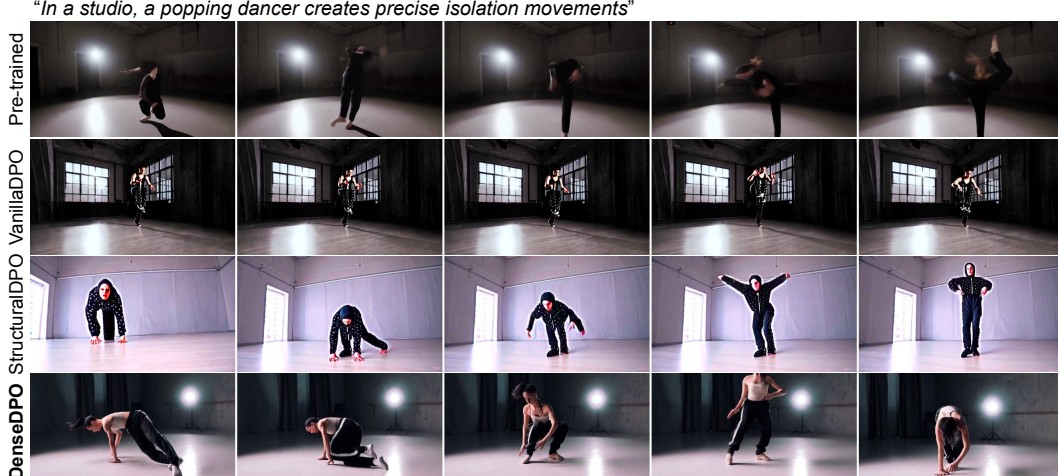

"*In a studio, a popping dancer creates precise isolation movements*"

Figure 4: **Qualitative results.** Pre-trained model generates deformed limbs. VanillaDPO fixes it but generates almost static motion. StructuralDPO retains dynamics but produces oversaturated frames. DenseDPO is the only method that generates correct limbs, large dynamics, and high quality visuals. **Please check out our project page for video results of baselines and our methods.**

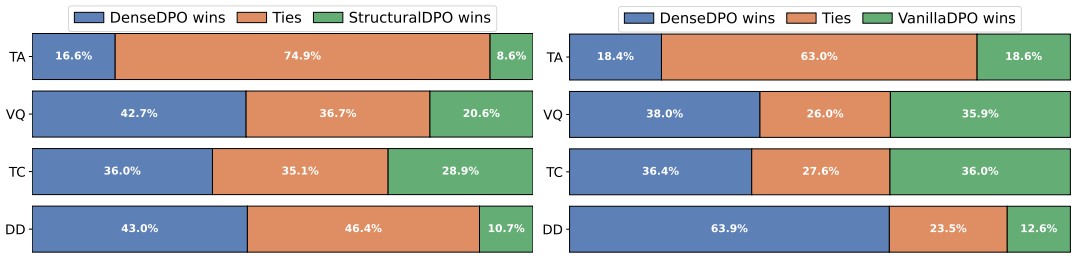

Figure 5: **Human evaluation** of DenseDPO *vs.* StructuralDPO (left) and VanillaDPO (right). TA, VQ, TC, DD stand for text alignment, visual quality, temporal consistency, and dynamic degree.

**Human evaluation.** We conduct a user study using all prompts from VideoJAM-bench in Fig. 5. We ask the participants to express their preference when presented with paired samples from our method and each baseline. DenseDPO consistently outperforms StructuralDPO in all dimensions. Compared to VanillaDPO, we achieve significantly higher dynamic degree, and are on par in other aspects. Please refer to Appendix C.2 for more user study results.

## 4.3 DPO with VLM Labels

**VLM labelers.** As discussed in Sec. 3.3, we aim to evaluate the effectiveness of pre-trained VLMs in video preference learning. We take VLMs designed specifically for the video quality assessment task, *VideoScore* [21], *LiFT* [74], *VideoReward* [46], and *VisionReward* [80]. They have been fine-tuned on large-scale human preference labels. We also utilize the state-of-the-art visual reasoning model, *GPT o3* [1], to explore the limits of models without task-specific training. Finally, we design *GPT o3 Segment* that partitions long videos into short segments to process separately, and aggregates results via majority voting. Please refer to Appendix A.6 for more implementation details.

**Evaluation setup.** We first test the preference prediction accuracy of VLMs on two types of videos: *Short Segment* partitions videos into 1s clips, and compares them separately. We report the accuracy on the 10k human preference labels used in DenseDPO. *Long Video* directly runs the model on the entire video except for GPT o3 Segment that aggregates segment-level results. We report the accuracy on the 30k human preference labels used in StructuralDPO. In addition, we conduct DPO training using these VLM-generated labels, and report the VisionReward score on VideoJAM-bench. Notably, we run VLMs to label video pairs generated from *all 55k* training data for better performance.

**Results.** Tab. 3a presents the results of preference prediction. With task-specific fine-tuning, state-of-the-art video reward models outperform the advanced GPT o3 in assessing short clips. Yet, their

Table 3: **VLMs results on human preference learning.** (a) Existing VLMs excel at assessing short segments, yet their performance on long videos is still unsatisfactory. In contrast, by aggregating results over short segments, GPT o3 achieves the best accuracy without task-specific fine-tuning. (b) DenseDPO training with GPT-based segment-level labels achieves performance close to using human labels, significantly outperforming other VLMs that predict single binary preferences.

| VLM | Short Segment | Long Video |
|---|---|---|
| VideoScore [21] | 61.23% | 51.23% |
| LiFT [74] | 65.03% | 55.45% |
| VideoReward [46] | 71.89% | 59.65% |
| VisionReward [80] | **72.45%** | 62.11% |
| GPT o3 [1] | 70.03% | 53.45% |
| GPT o3 Segment | 70.03% | **70.59%** |

(a) Preference accuracy of VLMs on 1s short segments and 5s long videos.

| Label Source | TA | VQ | TC | DD |
|---|---|---|---|---|
| Pre-trained | 0.770 | 0.192 | 0.354 | **0.680** |
| VisionReward [80] + StructuralDPO | 0.785 | 0.347 | 0.521 | 0.622 |
| GPT o3 [1] + StructuralDPO | 0.759 | 0.284 | 0.506 | 0.621 |
| GPT o3 Segment + DenseDPO | 0.842 | 0.368 | 0.598 | 0.672 |
| Human Label | **0.863** | **0.376** | **0.632** | 0.680 |

(b) DPO results with VLM labels on VideoJAM-bench [8].

Table 4: **Ablation on different variants of preference labels used for DPO training.** We report VBench [30] and VisionReward [80] metrics on VideoJAM-bench [8].

| Method | VBench Metrics | | | | | | VisionReward Metrics | | | |
|---|---|---|---|---|---|---|---|---|---|---|
| | Aesthetic Quality | Imaging Quality | Subject Consistency | Background Consistency | Motion Smoothness | Dynamic Degree | Text Alignment | Visual Quality | Temporal Consistency | Dynamic Degree |
| Pre-trained | 54.65 | 55.85 | 88.29 | 91.50 | 92.40 | 84.16 | 0.770 | 0.192 | 0.354 | 0.680 |
| StructuralDPO | 56.38 | 59.78 | 90.21 | 92.34 | 92.94 | 84.69 | 0.843 | 0.341 | 0.602 | 0.652 |
| Majority Voting | 56.48 | 59.72 | 90.15 | 92.35 | 92.98 | 84.54 | 0.847 | 0.342 | 0.608 | 0.645 |
| Flip 40% Label | 55.10 | 55.42 | 88.85 | 92.04 | 92.60 | 84.27 | 0.782 | 0.250 | 0.556 | 0.668 |
| Flip 20% Label | 56.23 | 58.10 | 90.85 | 93.06 | 92.85 | 84.85 | 0.846 | 0.368 | 0.602 | 0.678 |
| 0.5× Label | 55.62 | 56.13 | 89.23 | 92.48 | 92.99 | 84.25 | 0.806 | 0.305 | 0.589 | 0.655 |
| 2× Label | **57.56** | **62.02** | **92.85** | **95.06** | **93.85** | **85.55** | **0.882** | **0.395** | **0.654** | **0.682** |
| New GT Video | 56.94 | 60.91 | 91.48 | 93.76 | 93.57 | 85.39 | 0.868 | 0.377 | 0.636 | 0.678 |
| **DenseDPO** | 56.99 | 60.92 | 91.54 | 93.84 | 93.56 | 85.38 | 0.863 | 0.376 | 0.632 | 0.680 |

performance on long videos degrades drastically. Our further analysis in Appendix C.3 reveals that these models might be biased towards video content rather than temporal motion. Thanks to the temporal alignment of video pairs, we can run GPT to compare short segments individually, and then aggregate the results. This leads to higher accuracy in long video preference prediction.

We further show the DPO alignment results in Tab. 3b. Due to a higher preference accuracy, DenseDPO with GPT o3 Segment labels outperforms StructuralDPO with binary preferences from other VLMs. It even matches DenseDPO trained with human labels on text alignment, visual quality, and dynamic degree. Yet, please note that GPT label has 5.5× more videos than human label.

## 4.4 Ablation Study

We study the effect of each component in DenseDPO. All results are evaluated on VideoJAM-bench.

**Human labeling bias.** We first study if there is a systematic bias in the annotation pipeline, e.g., labelers may produce higher quality preference labels on short segments than long videos. To verify it, we aggregate labels of all segments within a video to obtain its binary preference label via *majority voting*, and then train StructuralDPO on these labels. Tab. 4 shows that it achieves similar results compared to StructuralDPO trained on globally labeled preferences. This proves that DenseDPO's superior performance comes from segment-level preference supervision instead of labeler bias.

**Human label quality.** We study how label noise affects DenseDPO performance. As shown in Tab. 4, we randomly flip *20%* and *40%* winning or losing labels. This results in a clear drop in all the metrics.

**Human label quantity.** We ablate different amounts of segment preference labels used in DenseDPO. Tab. 4 shows that scaling to *2×* labels leads to the best results across all metrics. Interestingly, results

Table 5: **Ablation on segment length $s$ of dense preference labels.** We report VBench [30] and VisionReward [80] metrics on VideoJAM-bench [8]. All models are only trained on 5k videos.

| Method | VBench Metrics | | | | | | VisionReward Metrics | | | |
|---|---|---|---|---|---|---|---|---|---|---|
| | Aesthetic Quality | Imaging Quality | Subject Consistency | Background Consistency | Motion Smoothness | Dynamic Degree | Text Alignment | Visual Quality | Temporal Consistency | Dynamic Degree |
| Pre-trained | 54.65 | 55.85 | 88.29 | 91.50 | 92.40 | 84.16 | 0.770 | 0.192 | 0.354 | 0.680 |
| $s = 0.5$ | 55.57 | **57.03** | **89.88** | **92.76** | 92.52 | 84.25 | **0.811** | **0.326** | **0.601** | 0.643 |
| $s = 1$ **(Ours)** | **55.62** | 56.13 | 89.23 | 92.48 | **92.99** | **84.25** | 0.806 | 0.305 | 0.589 | **0.655** |
| $s = 2$ | 55.40 | 56.10 | 89.04 | 91.96 | 92.54 | 84.15 | 0.795 | 0.291 | 0.557 | 0.623 |

Table 6: **Ablation on DenseDPO training with different amount of GPT o3 segment-level labels.** We report VBench [30] and VisionReward [80] metrics on VideoJAM-bench [8].

| Method | VBench Metrics | | | | | | VisionReward Metrics | | | |
|---|---|---|---|---|---|---|---|---|---|---|
| | Aesthetic Quality | Imaging Quality | Subject Consistency | Background Consistency | Motion Smoothness | Dynamic Degree | Text Alignment | Visual Quality | Temporal Consistency | Dynamic Degree |
| Pre-trained | 54.65 | 55.85 | 88.29 | 91.50 | 92.40 | 84.16 | 0.770 | 0.192 | 0.354 | 0.680 |
| 10k GPT labels | 55.21 | 56.83 | 88.31 | 91.45 | 92.38 | 84.20 | 0.782 | 0.247 | 0.360 | 0.668 |
| 35k GPT labels | 55.89 | 58.75 | 89.68 | 92.25 | 92.70 | 84.89 | 0.817 | 0.316 | 0.498 | 0.670 |
| 55k GPT labels | 56.23 | 60.15 | 90.75 | 93.01 | 92.99 | 85.21 | 0.842 | 0.368 | 0.598 | 0.672 |

trained with $0.5\times$ labels are clearly better than results trained with 40% labels flipped. This may indicate that the quality of labels has a larger impact than the quantity of labels.

**Ground-truth video in guided generation.** In our experiments, we randomly sample 10k videos from the 55k high-quality dataset to serve as guidance videos. Here, we randomly sample another set of 10k videos without overlap with the previously selected ones and conduct DenseDPO training. Tab. 4 (*New GT Video* row) shows that both runs achieve similar results, proving that our method is robust to the selection of ground-truth data as long as they are high-quality videos.

**Dense label granularity.** We study the impact of the segment length $s$ in dense preference labels. By default, $s = 1$ is used in all our experiments. Here, we tested $s = 0.5$ and $s = 2$. Due to the high annotation cost, we only label 5k videos for each segment setting in this study. Tab. 5 compares DenseDPO trained on 5k videos using different $s$. $s = 1$ consistently outperforms $s = 2$ due to more fine-grained preference annotation. Interestingly, $s = 0.5$ performs similarly to $s = 1$. We hypothesize that this is because $0.5s$ is too short—a longer context window is needed to assess the temporal aspect of videos. In addition, labeling dense preference at $s = 0.5$ is $2\times$ expensive compared to $s = 1$. Therefore, we choose to label 1s segments in our experiments.

**VLM label quantity.** Finally, we study DenseDPO performance when scaling up automatic VLM labels in Tab. 6. Since VLM labels are noisy, DenseDPO with 10k GPT labels achieves only marginal improvement compared to the pre-trained model. Nevertheless, as the amount of VLM labels grows, we see consistent improvement in all metrics. This trend reveals that the quantity of VLM labels compensates for their quality, and our method scales well with the quantity of automatic labels. Since VLM itself is an actively evolving field, we view this as a promising future direction to further scale up automatic preference learning with more data and better off-the-shelf VLMs.

## 5 Conclusion

We present DenseDPO, an improved preference optimization framework for video generation. We address two critical aspects of video DPO—comparison data curation and preference labeling. Our guided video generation mechanism and fine-grained preference labeling significantly improve over vanilla DPO. Furthermore, we show that segment-level labels unlock automatic annotation with off-the-shelf VLMs without task-specific training. We discuss our limitations in Appendix D.

**Acknowledgments**

We would like to thank Zhengyang Liang, Weize Chen, and Tsai-Shien Chen for valuable discussions, and Maryna Diakonova for help with data preparation and user studies.

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

# A  Detailed Experimental Setup

In this section, we provide full details on the datasets, baselines, evaluation settings, and the training and inference implementation details of our model.

## A.1  Training Data

Following common practice in diffusion model post-training [14, 35, 71], we curate a high-quality data subset from existing large-scale video datasets [12, 81]. We mostly follow [58] to filter the resolution, duration, aesthetic quality, and motion strength of videos. Inspired by VideoAlign [46], we further apply GPT-4o [3] to only retain prompts with non-trivial motion.

---

**GPT-4o Prompt Filtering Template**

Please help me classify if a text prompt contains challenging dynamics. Ignore camera motion and description of the background in the text prompt. Only focus on foreground objects. In general, we are looking for complex motion that requires precise, coordinated movement such as doing sports.

Here are some good examples:
1. A skateboarder performs jumps.
2. On a rainy rooftop, a pair of hip-hop dancers lock and pop in perfect sync.
3. A figure skater executes a powerful leap.
4. A woman transitions gracefully on an aerial hoop under golden hour light.
5. An acrobat holding a handstand on a mat.
6. A martial artist performs a spinning hook kick in a misty bamboo forest.

Additional rules:
1. Only keep real-time videos, and remove any video that is slow-motion, time-lapse, or aerial shot;
2. Remove videos with more than five major subjects in the scene, such as sports games or a group of people doing something;
3. Remove videos with any of the following content: screen-in-screen, screen recording, special effects from video editing, cartoon, animation, TV news, and video games;
4. When not violating the previous rules, you can keep videos with any of the following content: eating, cutting, or any action that causes big deformation of the main object, such as "A person takes a bite of a hamburger / cuts a steak / squeezes a sponge / makes a dough";
5. Be conservative. If you are uncertain about a prompt, please classify it as "no".

Please reply "yes" or "no" in the first line of your response. Then, please explain your decision in the second line. I will now provide the prompt for you to classify:
`[PROMPT]`

---

The GPT-4o template we used is shown above. Overall, our final dataset contains around 55k high-quality text-video pairs, which we used to generate preference data.

**Data processing.** The training dataset contains videos of different lengths, resolutions, and aspect ratios. Following common practice [57, 95], we use data bucketing, which groups videos into a fixed set of sizes. Overall, we sample videos up to 512 resolution and 5s during training.

## A.2  Baselines

**Pre-trained model.** Our base text-to-video model is a latent Diffusion Transformer framework [55]. It leverages a MAGVIT-v2 [85] as the autoencoder and a stack of 32 DiT blocks as the denoiser $\mathbf{G}_{\theta}$. The autoencoder is similar to the one in CogVideoX [84], which downsamples the spatial dimensions by $8\times$ and the temporal dimension by $4\times$. Each DiT block is similar to the one in Open-Sora [36], which consists of a 3D self-attention layer running on all video tokens, a cross-attention layer between video tokens and T5 text embeddings [62] of the input prompt, and an MLP.
The base model adopts the rectified flow training objective [44, 48]. We mostly follow the design choice of Stable Diffusion 3 [16], e.g., Logit-Normal distribution of noise levels.

**REFL** [79] directly maximizes the reward from a reward model via gradient ascent. We tried REFL in our preliminary experiments with HPSv2 [75] and VisionReward [80] as the target reward models. However, we observed severe reward hacking, e.g., the video frames become oversaturated and the

temporal motion gets static. It also requires huge GPU memory and computation as we need to decode latent tokens to raw pixels to apply these VLMs and backpropagate gradients through them. Since the results are clearly worse than DPO, we did not continue this direction and do not report their results in the paper.

**SFT** fine-tunes the pre-trained model on the 55k high-quality dataset described in Sec. A.1 for 5k iterations. We did not observe a clear difference between full-model and LoRA [27] fine-tuning, and thus choose LoRA fine-tuning as it is more efficient.

**D3PO** [83] shares a similar objective function as VanillaDPO. We adapt their official codebase[1] to our video setting, and use the same preference labels as in VanillaDPO.

**DPOK** [17] is an online RL-based method. We also adapt their official codebase[2] to our video setting, and leverage VisionReward [80] to provide online reward feedback. In our experiments, we found that DPOK is unstable to train and easily results in reward hacking. Thus, we use a low learning rate of $2 \times 10^{-6}$ and a high KL weight to prevent training divergence.

**SePPO** [90] can be viewed as a combination of DPO and SFT, as its preference data come from samples generated by a past checkpoint of the current model and pre-collected high-quality data. Since no training code is available for SePPO, we re-implement it based on its paper.

**VanillaDPO** follows the common direct preference optimization (DPO) practice for video diffusion models [46, 65, 80]. We randomly select 30k text prompts from the curated dataset, and generate 2 videos of 5s per prompt to obtain a comparison pair. We then ask human annotators to choose a better video or a tie by considering text alignment, visual quality, and temporal consistency. 2 labelers are assigned to a pair, with a reviewer to correct any potential errors. This leads to around 10k winning-losing pairs after removing ties. Indeed, human preferences in video are influenced by multiple, sometimes inversely correlated, factors, making it hard to obtain a clear preference.

**StructuralDPO** uses the same 30k prompts as VanillaDPO to construct comparison pairs. For a text prompt, we sample a guidance level $\eta \sim \mathcal{U}(0.65, 0.8)$, and 2 Gaussian noises from 2 random seeds. Then, we add the noise to the ground-truth video corresponding to the text prompt according to $\eta$, and denoise from it to generate a video. We perform the same human preference annotation process, again leading to around 10k non-tie data pairs.

We have tried using different $\eta$ when generating the video pair. Usually, the video generated with more guidance is preferred over the other one (e.g., has fewer artifacts). However, we observe that the model quickly achieves a low DPO loss when trained on this data, yet the quality of generated videos is not improved. We hypothesize that different $\eta$ leads to statistical differences in generated videos, and the model learns shortcuts using such cues instead of truly understanding human preferences.

**DPO training hyper-parameters.** Both VanillaDPO and StructuralDPO adopt the Flow-DPO loss with a constant $\beta$, as prior work shows it achieves the best performance [46]. Following prior works [46, 80], we use $\beta = 500$ and apply LoRA [27] with rank 128 to fine-tune the video model. We train with the AdamW optimizer [49] and a global batch size of 256 for 1k steps. The peak learning rate is set to $1 \times 10^{-5}$ and it is linearly warmed up from 0 in the first 250 steps. A gradient clipping of 1.0 is applied to stabilize training. We implement all models using PyTorch [54] and conduct training on 64 NVIDIA A100 GPUs, which takes around 16 hours.

**Inference hyper-parameters.** We use the rectified flow sampler [48] with 40 sampling steps and a classifier-free guidance [22] scale of 8 to generate horizontal videos of $512 \times 288$ resolution. A timestep shifting [16] of 5.66 is applied to further improve results.

### A.3 DenseDPO Implementation Details

For a fair comparison with baselines, we only label dense human preferences on 10k randomly sampled video pairs from the StructuralDPO training data, which costs a similar amount of human annotation time compared to labeling 30k global binary preferences. The segment length $s$ is set to 1s. Overall, more than 80% of video pairs have at least 1 non-tie segment and can be used in DPO training, greatly improving the data efficiency over using global preferences. Since the DPO loss is now averaged over all non-tie tokens within a training video pair, we sample video clips that have

---

[1] https://github.com/yk7333/d3po
[2] https://github.com/google-research/google-research/tree/master/dpok

more than 20% non-tie segments to avoid a small effective batch size. All other training and inference hyper-parameters are the same as DPO baselines.

## A.4 Evaluation Datasets

At the beginning of the project, we experimented with the prompt suite of VBench [30]. However, we noticed that the model can achieve high scores with good visual quality and even degraded motion dynamics. Since our main focus is improving the temporal quality of the pre-trained video model and VanillaDPO, we choose to evaluate on two motion-rich prompt sets.

**VideoJAM-bench** [8] contains 128 prompts focusing on real-world scenarios with challenging motion, ranging from human actions to physical phenomena.

**MotionBench** collects more diverse prompts from existing prompt sets [35, 58, 74]. We run GPT-4o to select prompts with dynamic actions as described in Sec. A.1, resulting in 419 prompts. This prompt set contains more diverse scenes, subjects, and action types.

Note that, to ensure a fair comparison between methods, we use the original text prompt without any prompt rewriting or extension process.

## A.5 Evaluation Metrics

Inspired by prior works [46, 80], we identify three key aspects in text-to-video generation: visual quality, text alignment, and motion quality. Since VanillaDPO has the motion bias issue that leads to visually pleasing videos with reduced motion strength, we want to evaluate both the smoothness (i.e., temporal consistency) and the strength (i.e., dynamic degree) of the video in a disentangled way. Thus, we cannot use metrics like VideoReward [46] as it only contains a "motion quality" dimension.

**VBench** [30] is a comprehensive benchmark that tests different aspects of a video generation model. When testing on custom prompt sets, it supports six dimensions covering the visual quality, temporal consistency, and dynamic degree. Following the official evaluation protocol, we run each model to generate 5 videos using each prompt with 5 random seeds.

**VisionReward** [80] is a state-of-the-art video reward model that fine-tunes a pre-trained Vision Language Model (VLM) [26] on large-scale human video preference data. It breaks down the human preference into 64 aspects, which can be categorized into 9 dimensions. We take the "Alignment" dimension as text alignment, merge "Composition", "Quality", "Fidelity" dimensions into visual quality, merge "Stability", "Physics", "Preservation" dimensions into temporal consistency, and take the "Dynamic" dimension as dynamic degree to report in the final result.

## A.6 VLM-based Preference Labeling

**VLM labelers.** We aim to evaluate the effectiveness of existing VLMs in video preference learning. This includes both off-the-shelf models and models fine-tuned for the preference prediction task. For fine-tuned VLMs, we input both videos and take the one with a higher score as the winning data:

- *VideoScore* [21] is fine-tuned on human-labeled scores on 1-3s short videos. We average all five dimensions of its output as the overall score of a video;

- *LiFT* [74], *VideoReward* [46], and *VisionReward* [80] are all fine-tuned on >5s long videos generated by modern video models. We average their output dimensions or take the overall dimension (if presented) as the final score of a video.

For off-the-shelf VLMs, we take *GPT o3* [1] as we find it to outperform GPT-4o [3] due to the visual reasoning capability. We follow the official guide[3] to prompt the model as follows:

---

[3]https://cookbook.openai.com/examples/gpt_with_vision_for_video_understanding

> **GPT o3 Video Preference Labeling Template**
>
> Please help me compare two videos generated by our text-to-video diffusion model. I will provide you with frames sampled from the two videos. The two videos are structurally similar (e.g. global layout and motion are similar), so I only want to compare their details. Please assess which one has a higher quality, i.e., the video that contains fewer artifacts. Pay close attention to visual artifacts such as:
> - Additional fingers or legs, deformed human limbs, morphing human faces or body parts;
> - Blurry or distorted objects, slight motion blur is fine, but the object should not be completely distorted;
> - Abrupt changes in the object, such as objects appearing/disappearing unexpectedly, or anything that should not happen in the real world, e.g., rigid object deforming or melting.
>
> Please only answer "tie" (two videos have equal quality), "first" (the first video has fewer artifacts), and "second" (the second video has fewer artifacts), followed by a simple explanation.
> Be conservative in your answer. If you see similar amounts of artifacts in both videos, please choose "tie". Only select "first" or "second" when one video is clearly better than the other.
>
> These are frames from the first video, sampled at 8 FPS:
> `Video 1 frames`
>
> These are frames from the second video, sampled at 8 FPS:
> `Video 2 frames`
>
> Please compare the two videos and tell me which one is better.

To improve accuracy, we apply a self-consistency check by reversing the order of `Video 1` and `Video 2`. If the predictions on both orders are the same, we keep it. Otherwise, we treat it as a tie. This simple strategy improves the accuracy on short segments by around 10%. We note that there might be better strategies in prompt construction, such as concatenating paired video frames side-by-side, or organizing frames into a grid. We leave further investigations for future work.

Finally, we design *GPT o3 Segment* that partitions long videos into short 1s segments to process separately. This gives the dense preference labels compatible with our DenseDPO framework. To obtain a global preference for the entire long video, we simply apply majority voting.

**Preference prediction setup.** Prior work [46] pointed out that existing VLMs excel at processing short videos, while falling short on long videos. Therefore, we evaluate two cases:

- *Short Segment* that predicts human preferences on 1s clips. We report the accuracy on the 10k dense human preference labels used in DenseDPO training;
- *Long Video* that predicts human preferences on the entire video, except for GPT o3 Segment that aggregates segment-level results. We report the accuracy on the 30k binary human preference labels used in StructuralDPO training, which is a superset of the previous case.

When calculating the prediction accuracy, we skip tie labels and only compute results on segments or videos with non-tie ground-truth preference labels.

**DPO training setup.** We apply StructuralDPO on binary preference labels produced by GPT o3 and VisionReward as it achieves the highest accuracy. We also apply DenseDPO on dense preference labels produced by GPT o3 Segment. Notably, we run VLMs to label video pairs generated from *all 55k* training data to explore the limit of automatic preference learning performance.

# B Caveat of Guided Sampling

## B.1 Analyzing the Learning Signal

As discussed in Sec. 3.2, guided sampling is attractive since it fixes the structure in the preference pair. This neutralizes the motion bias between videos and focuses the comparison on visual artifacts. However, StructuralDPO with guided sampling achieves inferior results. Analyzing its learning signal reveals that it can paradoxically push the model to "*unlearn*" the real data distribution. Intuitively, this happens because the learning signal from a losing sample typically dominates over the winning one in those regions of a video, which correspond to the real data distribution. In this section, we investigate this phenomenon and discuss potential remedies to the issue.

For our analysis, we will use the original DPO notation [70] to simplify the exposition and emphasize that the argument applies for the most general diffusion DPO setup. Diffusion DPO training loss is formulated as:

$$\mathcal{L}(\theta) = -\mathbb{E}_{(\boldsymbol{x}_0^w, \boldsymbol{x}_0^l) \sim \mathcal{D}, \, t \sim \mathcal{U}(0,T), \, \boldsymbol{x}_t^w \sim q(\boldsymbol{x}_t^w | \boldsymbol{x}_0^w), \, \boldsymbol{x}_t^l \sim q(\boldsymbol{x}_t^l | \boldsymbol{x}_0^l)} \Big[ \log \sigma \Big( - \beta T \big( \tag{8}$$

$$+ \mathbb{D}_{\mathrm{KL}} \big( q(\boldsymbol{x}_{t-1}^w \mid \boldsymbol{x}_{0,t}^w) \,\|\, p_\theta(\boldsymbol{x}_{t-1}^w \mid \boldsymbol{x}_t^w) \big) - \mathbb{D}_{\mathrm{KL}} \big( q(\boldsymbol{x}_{t-1}^w \mid \boldsymbol{x}_{0,t}^w) \,\|\, p_{\mathrm{ref}}(\boldsymbol{x}_{t-1}^w \mid \boldsymbol{x}_t^w) \big) \tag{9}$$

$$- \mathbb{D}_{\mathrm{KL}} \big( q(\boldsymbol{x}_{t-1}^l \mid \boldsymbol{x}_{0,t}^l) \,\|\, p_\theta(\boldsymbol{x}_{t-1}^l \mid \boldsymbol{x}_t^l) \big) + \mathbb{D}_{\mathrm{KL}} \big( q(\boldsymbol{x}_{t-1}^l \mid \boldsymbol{x}_{0,t}^l) \,\|\, p_{\mathrm{ref}}(\boldsymbol{x}_{t-1}^l \mid \boldsymbol{x}_t^l) \big) \big) \Big) \Big], \tag{10}$$

where $(\boldsymbol{x}_0^w, \boldsymbol{x}_0^l)$ is the winning-losing preference pair, $T$ is the number of diffusion steps, $t \sim \mathcal{U}(0,T)$ is the noise level distribution, and $q(\boldsymbol{x}_t \mid \boldsymbol{x}_0) = \mathcal{N}(\boldsymbol{x}_t \mid \alpha_t \boldsymbol{x}_0, \sigma_t I)$. In Diffusion-DPO [70], the objective is further simplified to:

$$\mathcal{L}(\theta) = -\mathbb{E}_{(\boldsymbol{x}_0^w, \boldsymbol{x}_0^l) \sim \mathcal{D}, \, t \sim \mathcal{U}(0,T), \, \boldsymbol{x}_t^w \sim q(\boldsymbol{x}_t^w | \boldsymbol{x}_0^w), \, \boldsymbol{x}_t^l \sim q(\boldsymbol{x}_t^l | \boldsymbol{x}_0^l)} \Big[ \log \sigma \Big( - \beta T \omega(t) \big( \tag{11}$$

$$\|\boldsymbol{\epsilon}^w - \boldsymbol{\epsilon}_\theta(\boldsymbol{x}_t^w, t)\|_2^2 - \|\boldsymbol{\epsilon}^w - \boldsymbol{\epsilon}_{\mathrm{ref}}(\boldsymbol{x}_t^w, t)\|_2^2 - \|\boldsymbol{\epsilon}^l - \boldsymbol{\epsilon}_\theta(\boldsymbol{x}_t^l, t)\|_2^2 + \|\boldsymbol{\epsilon}^l - \boldsymbol{\epsilon}_{\mathrm{ref}}(\boldsymbol{x}_t^l, t)\|_2^2 \big) \Big) \Big], \tag{12}$$

where $\omega(t)$ is a weighting function. Let's denote:

$$\Delta_\theta^w = \|\boldsymbol{\epsilon}^w - \boldsymbol{\epsilon}_\theta(\boldsymbol{x}_t^w, t)\|_2^2 \qquad \Delta_\theta^l = \|\boldsymbol{\epsilon}^l - \boldsymbol{\epsilon}_\theta(\boldsymbol{x}_t^l, t)\|_2^2 \tag{13}$$

$$\Delta_{\mathrm{ref}}^w = \|\boldsymbol{\epsilon}^w - \boldsymbol{\epsilon}_{\mathrm{ref}}(\boldsymbol{x}_t^w, t)\|_2^2 \qquad \Delta_{\mathrm{ref}}^l = \|\boldsymbol{\epsilon}^l - \boldsymbol{\epsilon}_{\mathrm{ref}}(\boldsymbol{x}_t^l, t)\|_2^2 \tag{14}$$

Then, the loss gradient is:

$$\nabla_\theta \mathcal{L}(\theta) = -\nabla_\theta \mathbb{E} \big[ \log \sigma \big( -\beta T \omega(t) \cdot \big( \Delta_\theta^w - \Delta_{\mathrm{ref}}^w - (\Delta_\theta^l - \Delta_{\mathrm{ref}}^l) \big) \big) \big] \tag{15}$$

$$= \mathbb{E} \big[ (1 - \sigma(\cdot)) \beta T \omega(t) \cdot \nabla_\theta (\Delta_\theta^w - \Delta_{\mathrm{ref}}^w - (\Delta_\theta^l - \Delta_{\mathrm{ref}}^l)) \big] \tag{16}$$

$$= \mathbb{E} \big[ C \cdot \nabla_\theta (\Delta_\theta^w - \Delta_\theta^l) \big], \tag{17}$$

where $C = (1 - \sigma(\cdot)) \beta T \omega(t)$.

One can show that $C > 0$ since $\sigma(\cdot) \in (0,1)$ and $\beta, T, \omega(t) > 0$. Since $C > 0$, the per-pixel direction of the incoming gradient w.r.t. the model outputs is determined entirely by the interplay between $\nabla_\theta \Delta_\theta^w$ and $\nabla_\theta \Delta_\theta^l$. This fact becomes crucial for StructuralDPO with guided sampling for the following reason.

In guided sampling (see Algo. 2), we generate a winning/losing sample $\boldsymbol{x}_n^* = (1-\eta)\boldsymbol{x}_0 + \eta \boldsymbol{\epsilon}^*$ using a real video $\boldsymbol{x}_0$. This makes them carry similar structure, which means that the winning and losing samples share many pixels. Moreover, these shared pixels normally correspond to the ground-truth data distribution. Let's split the pixels into two sets, $\mathcal{I}_{\mathrm{same}}$ and $\mathcal{I}_{\mathrm{unique}}$:

$$\mathcal{I}_{\mathrm{same}}(\boldsymbol{x}^w, \boldsymbol{x}^l) = \{ p \mid \boldsymbol{x}_0^w[p] \approx \boldsymbol{x}_0^l[p] \approx \boldsymbol{x}_0[p] \}, \tag{18}$$

$$\mathcal{I}_{\mathrm{unique}}(\boldsymbol{x}_0^w, \boldsymbol{x}_0^l) = \{ p \mid p \notin \mathcal{I}_{\mathrm{same}} \}, \tag{19}$$

where $p$ is a pixel location (e.g., typically, triplet $(i, j, k)$ denoting the frame, height, width indices). In this way, $\mathcal{I}_{\mathrm{same}}$ stores pixel locations which remained intact during the forward diffusion process of guided sampling and subsequent denoising with $\mathbf{G}_{\boldsymbol{\theta}}$. Fig. 2 and Fig. 11 illustrate this: many pixels in the winning and losing samples are identical and correspond to the original ground-truth video. In this way, one can argue that $\mathcal{I}_{\mathrm{same}}$ corresponds to the real data distribution.

Let's denote:

$$\Delta_{(\cdot)}^*[\mathcal{I}] = \|\boldsymbol{\epsilon}^*[\mathcal{I}] - \boldsymbol{\epsilon}_{(\cdot)}^*(\boldsymbol{x}_t^*, t)[\mathcal{I}]\|_2^2, \tag{20}$$

i.e., the diffusion loss in particular pixel locations $\mathcal{I}$. Now, if $\Delta_\theta^w[\mathcal{I}_{\mathrm{same}}] < \Delta_\theta^l[\mathcal{I}_{\mathrm{same}}]$ (meaning that the diffusion error in ground-truth pixel locations of a losing sample is higher than that of the winning one), then the gradient will be dominated by the negative contribution of $\Delta_\theta^l[\mathcal{I}_{\mathrm{same}}]$, and the model will be *unlearning* real data distribution. Turns out, this is exactly what is happening in practice:

- Prior to DPO, the model undergoes extensive supervised fine-tuning (SFT), so it is reasonable to expect that at initialization, we have $\mathbb{D}_{\mathrm{KL}} \big( q(\boldsymbol{x}_{t-1}^w \mid \boldsymbol{x}_{0,t}^w) \,\|\, p_\theta(\boldsymbol{x}_{t-1}^w \mid \boldsymbol{x}_t^w) \big) <$

$\mathbb{D}_{\mathrm{KL}}\left(q(\boldsymbol{x}_{t-1}^{l} \mid \boldsymbol{x}_{0,t}^{l}) \,\|\, p_{\theta}(\boldsymbol{x}_{t-1}^{l} \mid \boldsymbol{x}_{t}^{l})\right)$, meaning that the model is closer in distribution to winning pairs and interprets the entire image as unlikely when there are artifacts present in some part of it. The condition $\mathbb{D}_{\mathrm{KL}}\left(q(\boldsymbol{x}_{t-1}^{w} \mid \boldsymbol{x}_{0,t}^{w}) \,\|\, p_{\theta}(\boldsymbol{x}_{t-1}^{w} \mid \boldsymbol{x}_{t}^{w})\right) <$ $\mathbb{D}_{\mathrm{KL}}\left(q(\boldsymbol{x}_{t-1}^{l} \mid \boldsymbol{x}_{0,t}^{l}) \,\|\, p_{\theta}(\boldsymbol{x}_{t-1}^{l} \mid \boldsymbol{x}_{t}^{l})\right)$ implies $\Delta_{\theta}^{w} < \Delta_{\theta}^{l}$ (see Appendix 2 of [70]).

- The model suffers from an internal distribution shift in the presence of artifacts and its predictions in good pixel locations deteriorate in the presence of artifacts. Besides, it might be shifting its capacity towards rectifying the artifacts, so the rest of the output suffers.

Given the above two observations, we conclude that the model will be unlearning the real data distribution in the StructuralDPO setting. While the above analysis is outlined for diffusion models with $\epsilon$-prediction parametrization [23], it holds for $v$-prediction as well (used in both our and many contemporary works that align rectified flow models [48]) with argument derivation being basically the same. We also emphasize that even marginal domination of the gradient from a losing sample over the one from the winning sample results in such "unlearning" behavior.

Moreover, practitioners commonly use the same noise seed (i.e., $\epsilon^{w} = \epsilon^{l}$) for both the winning and losing samples, following the original DiffusionDPO implementation.[4] This exacerbates the problem: the learning signal from a losing sample no longer dominates merely in expectation per pixel, but at each training step, entirely suppressing the learning signal from the winning sample.

Several prior works observed a similar behavior in DPO on language models [18, 53, 63]. For example, DPO-Positive [53] shows that on datasets with short edit distances between winning and losing samples, DPO may lead to a reduction in the model's likelihood on the preferred examples.

DenseDPO is a natural way to eliminate this shortcoming of StructuralDPO since it is designed to provide dense per-pixel DPO objective and would allow to treat the pixels from $\mathcal{I}_{\mathrm{same}}$ (i.e., similar pixels) as ties, thus removing any loss on them. In the ideal world, we would love to have per-pixel preference labels for DenseDPO training, but, as our work demonstrates, even coarse-level temporal ones allow us to recover and improve the DPO performance.

One can also investigate other strategies to mitigate the issue by taking into account the similarity between pixels (e.g., uncertainty-based or margin-aware DPO [25, 47]) or reformulating the DPO in some novel way without maximizing the loss of losing samples. We leave this for future work.

## B.2 Empirical Verification

While the previous section presents our theoretical argument, we must verify empirically whether the assumption regarding the dominance of the losing sample's loss over the winning sample indeed holds. Here, we address this question through empirical analysis using the Flux-dev [37] model. We deliberately chose a popular, open-source $v$-prediction model to ensure our conclusions remain general and reproducible rather than specific to our internal video model.

We constructed a synthetic dataset containing controlled artifacts to facilitate this analysis. Specifically, we selected $5,195$ real-world images with a resolution of $1024^2$ and artificially corrupted them by applying blur to their central patches. Each image is encoded using the FluxAE encoder, resulting in a latent tensor of dimensions $128 \times 128 \times 16$. We then applied increasing levels of corruption to the central $32 \times 32$ patch (representing 6.25% of the image) with the following blur intensities:

1. No corruption (0% blur)
2. Blur with $k = 2$ (6.25% of the patch size)
3. Blur with $k = 4$ (12.5% of the patch size)
4. Blur with $k = 8$ (25% of the patch size)
5. Blur with $k = 16$ (50% of the patch size)

The resulting dataset comprises five image variants, progressively more corrupted, visualized in Fig. 6 (top). This setup allows separate measurement of losses in corrupted and uncorrupted regions. Subsequently, we randomly sample a timestep $t \sim \mathcal{U}[0,1]$, estimate velocities $\boldsymbol{v}_{\mathrm{Flux}}(\boldsymbol{x}_{t}^{w}, t)$ and

---

[4] https://github.com/SalesforceAIResearch/DiffusionDPO/blob/main/train.py#L1053-L1055

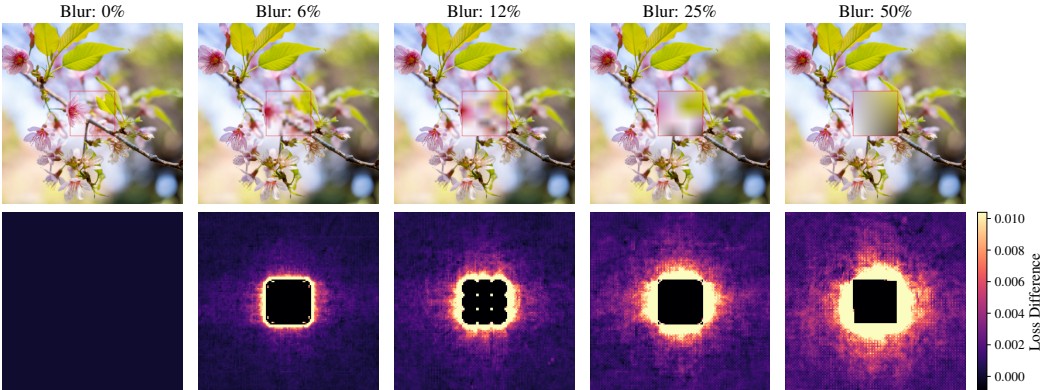

Figure 6: **Visualization of losing sample loss dominance in *uncorrupted regions*.** Top: example images with progressively increasing blur in the central region (note, that the visualization corresponds to an equally-corrupted RGB image, rather than the decoded corrupted latent tensor). Bottom: the per-pixel loss difference $\delta_{\mathcal{L}}$ between losing and winning samples averaged across latent channels. Positive values indicate the dominance of losing sample losses, driving the model to unintentionally degrade predictions in artifact-free areas. For clarity, we clamp maximum values in the visualization to the maximum loss difference observed in uncorrupted regions, preventing extreme outliers from dominating the heatmap.

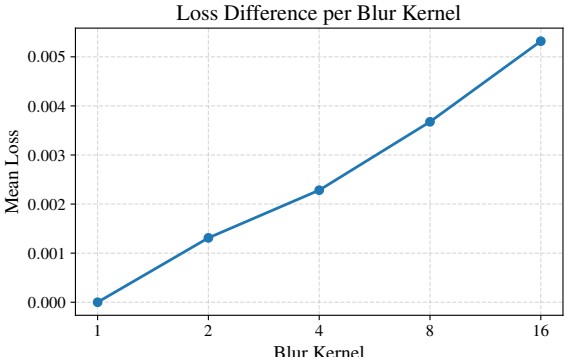

Figure 7: Average loss difference $\texttt{mean}(\delta_{\mathcal{L}})$ in *uncorrupted regions* as a function of artifact severity. Increasing blur severity amplifies the losing sample's negative learning signal in good (uncorrupted) regions, illustrating the risk of inadvertently unlearning correct predictions.

$\boldsymbol{v}_{\text{Flux}}(\boldsymbol{x}_t^l, t)$, and compute per-pixel losses (averaged over the 16 latent channels $C$):

$$\boldsymbol{\mathcal{L}}^w = \frac{1}{C} \sum_{c=1}^{C} (\boldsymbol{v}_{\text{Flux}}(\boldsymbol{x}_t^w, t)[c] - (\boldsymbol{\epsilon} - \boldsymbol{x}_0^w)[c])^2 \tag{21}$$

$$\boldsymbol{\mathcal{L}}^l = \frac{1}{C} \sum_{c=1}^{C} (\boldsymbol{v}_{\text{Flux}}(\boldsymbol{x}_t^l, t)[c] - (\boldsymbol{\epsilon} - \boldsymbol{x}_0^l)[c])^2, \tag{22}$$

where $(\cdot)^2$ denotes element-wise squaring. Next, for each sample, we calculate the loss difference $\delta_{\mathcal{L}} \in \mathbb{R}^{h \times w}$ as follows:

$$\delta_{\mathcal{L}} = \boldsymbol{\mathcal{L}}^l - \boldsymbol{\mathcal{L}}^w. \tag{23}$$

$\delta_{\mathcal{L}}$ indicates the extent to which the losing sample's loss surpasses that of the winning sample. If this dominance occurs in $\mathcal{I}_{\text{same}}$, it implies the model is *unlearning* these areas due to the negative contribution from the losing sample.

Loss computations use the same noise seed for the corrupted and uncorrupted images (as is usually done in practice). Fig. 6 (top) illustrates a representative example of corruption. Visualizations in Fig. 6 (bottom) clearly demonstrate the loss dominance of the losing samples in the *uncorrupted*

Table 7: **Quantitative comparison with more baselines on VideoJAM-bench [8].** We report automatic metrics from VBench [30] and VisionReward [80].

| Method | VBench Metrics | | | | | | VisionReward Metrics | | | |
| --- | --- | --- | --- | --- | --- | --- | --- | --- | --- | --- |
| | Aesthetic Quality | Imaging Quality | Subject Consistency | Background Consistency | Motion Smoothness | Dynamic Degree | Text Alignment | Visual Quality | Temporal Consistency | Dynamic Degree |
| Pre-trained | 54.65 | 55.85 | 88.29 | 91.50 | 92.40 | 84.16 | 0.770 | 0.192 | 0.354 | **0.680** |
| D3PO [83] | 56.15 | 58.03 | 88.93 | 92.23 | 92.78 | 82.53 | 0.833 | 0.322 | 0.482 | 0.602 |
| DPOK [17] | 54.99 | 56.28 | 89.14 | 92.36 | 92.95 | 78.65 | 0.795 | 0.337 | 0.518 | 0.457 |
| SePPO [90] | 55.83 | 57.42 | 89.93 | 92.65 | 92.93 | 82.85 | 0.841 | 0.326 | 0.554 | 0.587 |
| Vanilla DPO [46] | **57.25** | 60.38 | 91.21 | **93.94** | 93.43 | 80.25 | **0.867** | 0.371 | **0.636** | 0.535 |
| **DenseDPO** | 56.99 | **60.92** | **91.54** | 93.84 | **93.56** | **85.38** | 0.863 | **0.376** | 0.632 | 0.680 |

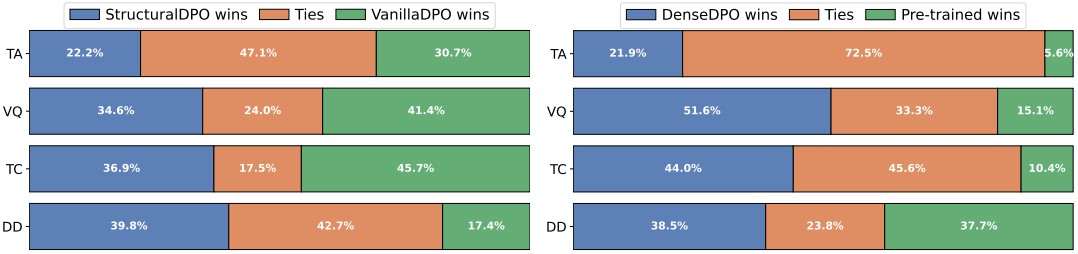

    (a) StructuralDPO *vs.* VanillaDPO            (b) DenseDPO *vs.* pre-trained model

Figure 8: **Human evaluation** on the VideoJAM-bench dataset. TA, VQ, TC, DD stand for text alignment, visual quality, temporal consistency, and dynamic degree.

*regions*. As we maximize the loss of the losing sample and minimize that of the winning sample, this results in the model increasing loss in uncorrupted image regions when trained with DPO to mitigate blurring artifacts.

Fig. 7 provides a quantitative analysis, demonstrating how increasing artifact severity intensifies the negative learning signal from losing samples in uncorrupted image areas.

## C Additional Experimental Results

### C.1 Comparison with Additional Baselines

In Tab. 7, we compare with additional baselines on VideoJAM-bench [8]. *D3PO* [83] achieves similar results as VanillaDPO since they are both DPO-based methods. *DPOK* [17], an online RL method, slightly improves the visual quality and temporal consistency compared to the pre-trained model, yet significantly degrades the dynamic degree. This is because the reward model it uses, VisionReward [80], is biased towards per-frame visual quality instead of temporal motion (see more analysis in C.3). Thus, training with such online feedback leads to static motion. *SePPO* [90] achieves a better dynamic degree than VanillaDPO, while performing worse in visual quality. Instead of using offline-generated losing samples, SePPO generates online samples with a reference model, and designs a filtering method (AAF) to assess its quality. However, we noticed that AFF is based on model denoising loss, which is sometimes unreliable. In those cases, the model is optimized on low-quality videos, leading to degraded visual quality.

### C.2 DPO with Human Labels

**Human evaluation.** Fig. 8 presents an additional user study on VideoJAM-bench. Fig. 8a shows that StructuralDPO outperforms VanillaDPO in dynamic degree as it performs DPO on video pairs with similar motion. Yet, it underperforms in other dimensions. Fig. 8b shows that DenseDPO consistently beats the pre-trained model in all dimensions.

**Qualitative results.** We show more videos generated by DenseDPO in Fig. 9. Fig. 10 compares DenseDPO with baselines. Overall, DenseDPO aligned model generates videos with high visual quality, rich motion, and precise text alignment. **Please check out our project page for video results of baselines and our methods.**

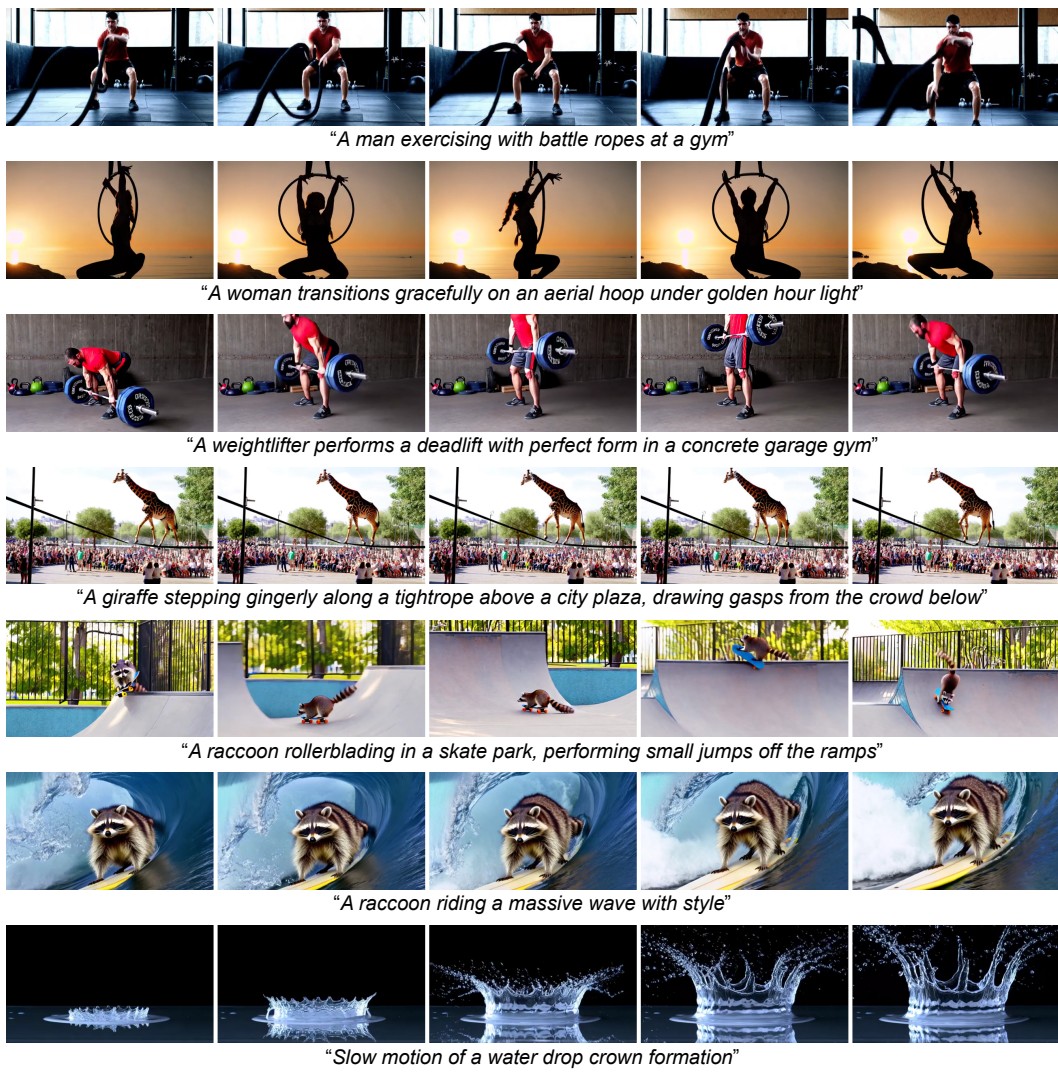

"A man exercising with battle ropes at a gym"

"A woman transitions gracefully on an aerial hoop under golden hour light"

"A weightlifter performs a deadlift with perfect form in a concrete garage gym"

"A giraffe stepping gingerly along a tightrope above a city plaza, drawing gasps from the crowd below"

"A raccoon rollerblading in a skate park, performing small jumps off the ramps"

"A raccoon riding a massive wave with style"

"Slow motion of a water drop crown formation"

Figure 9: **Text-to-video results with our DenseDPO aligned model.** Here, we show generation of challenging human activities, novel animal actions, and physical phenomena. **Please check out our project page for video results of baselines and our methods.**

## C.3  DPO with VLM Labels

**Motion bias in VLMs.** As discussed in Sec. 3.2, there is a motion bias in human preference annotation—human labelers tend to favor artifact-free slow-motion clips over dynamic clips with artifacts. Ge et al. [20] pointed out that video metrics such as FVD [69] are also biased towards per-frame visual quality rather than temporal motion quality. Here, we study whether more advanced VLM-based video reward models also suffer from this issue. We test two state-of-the-art models, VisionReward [80] and VideoReward [46]. We randomly sample 10k videos from the VanillaDPO training data, each having 121 frames (5s). For each video, we construct a static version of it by duplicating one frame of it, and we compare this static video with the original video.

Tab. 8 presents the winning rate of static *vs.* original video. Surprisingly, VisionReward favors static videos over original videos with a sizable gap, indicating a clear motion bias. In contrast, VideoReward prefers original videos in around 80% of cases. We note that both VisionReward and VideoReward output multi-dimensional scores (e.g., visual quality, dynamic degree), and aggregate them to predict the binary human preference. VideoReward simply averages all dimensions, and thus is not biased to any dimension. VisionReward instead first labels human preferences on video pairs, and then learns per-dimension weights via logistic regression. This inevitably inherits the motion

Table 8: **Winning rate of static video *vs.* original video measured by video reward models.** The static video is constructed by duplicating a frame to the video length, where we tested frame 0, 24, 48, and 96 here. A lower winning rate means the video reward model is more sensitive to motion.

| Method | Frame 0 *vs.* Original video | Frame 24 *vs.* Original video | Frame 48 *vs.* Original video | Frame 96 *vs.* Original video |
|---|---|---|---|---|
| VisionReward [80] | 70.63% | 68.28% | 69.84% | 69.06% |
| VideoReward [46] | 20.33% | 17.76% | 17.72% | 17.91% |

bias in human labels. Indeed, Tab. 25 in the VisionReward paper [80] reveals that human preference is *negatively* correlated with object dynamics, while *positively* correlated with temporal smoothness.

These results suggest that it is better to label per-dimension scores and predict them, instead of predicting an overall score. The bias in human preferences will be leaked into the reward model if we train the model to regress it. We note that our analysis is still preliminary. Further investigations similar to [20] are required to fully understand the bias in recent VLM-based video reward models.

**GPT dense preference label.** We visualize some dense preferences predicted by GPT o3 Segment in Fig. 11. Overall, GPT is able to identify obvious artifacts such as distorted faces and deformed limbs. With our carefully designed prompt and self-consistency check, it only predicts a preference when there is a clear difference between two segments. However, it still does not understand complex motion, such as playing tennis and cartwheels. This is partially because GPT o3 Segment only has access to 1s video clips, which is too short to finish these actions.

## C.4 Failed Attempts

Here, we record some failed approaches we tried.

**Real videos as winning data.** Since our goal is to improve the motion quality of video models, real-world videos can naturally serve as winning samples as they follow physical laws perfectly. We tried DPO using our 55k high-quality training videos as winning data, and videos generated by our model as losing data. The implicit accuracy quickly converges to almost 100% within 500 steps, yet the model generation does not improve. This is likely because real videos are significantly better than the generated ones, which fails to provide useful signals to improve the video model.

**Video pairs with different guidance $\eta$.** Intuitively, videos generated with more guidance are often better than those with lower guidance. We tried generating video pairs with different $\eta$ and assign winning-losing labels based on it. This gives us preference labels "for-free". However, DPO on this data does not improve model generation. Our hypothesis is that the model may infer the noise level from the generated samples instead of measuring its quality.

## D Limitations and Future Works

Similar to prior works [46, 80], we also observed unstable training and reward hacking when fine-tuning the entire model. As a result, we have to rely on LoRA training and early stopping. This is in stark contrast to DPO in large language models, where DPO training is relatively stable. More investigation on diffusion DPO basics is needed to resolve this issue.

To mitigate the motion bias in VanillaDPO, we propose guided sampling [51] to generate structurally-similar pairs. However, this reduces the variations in comparison pairs, degrading the DPO performance. We note that image-to-video generation with the same conditioning image is another way to retain similar structure between video pairs. In addition, it allows more variations in the generated videos, which may improve StructuralDPO performance.

Finally, our segment-level preference optimization method can be useful beyond the DPO framework. Our experiments show that even SOTA video reward models are bad at assessing all aspects of a generated video, yet effective online RL training relies on good reward feedback [45, 82]. An interesting direction is to leverage VLM to provide higher-quality dense reward feedback, and adapt RL methods to optimize such dense reward signals.

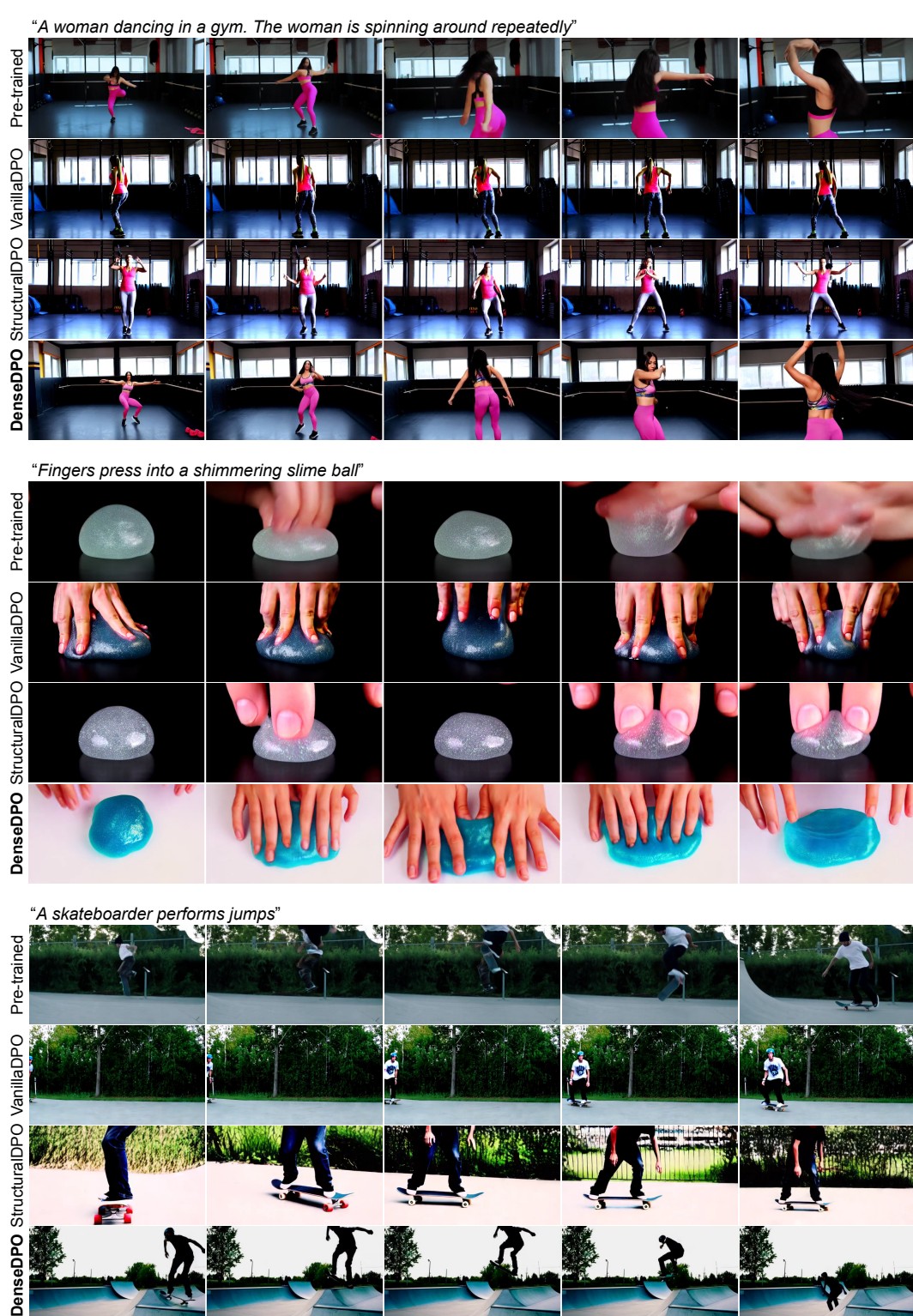

Figure 10: **Qualitative comparison with baselines.** Pre-trained model often generates deformed human body or unnatural object composition. VanillaDPO fixes these artifacts, but with significantly reduced dynamics. StructuralDPO retains the dynamics, but generates oversaturated frames or some artifacts. DenseDPO strikes the best balance over these dimensions. **Please check out our project page for video results of baselines and our methods.**

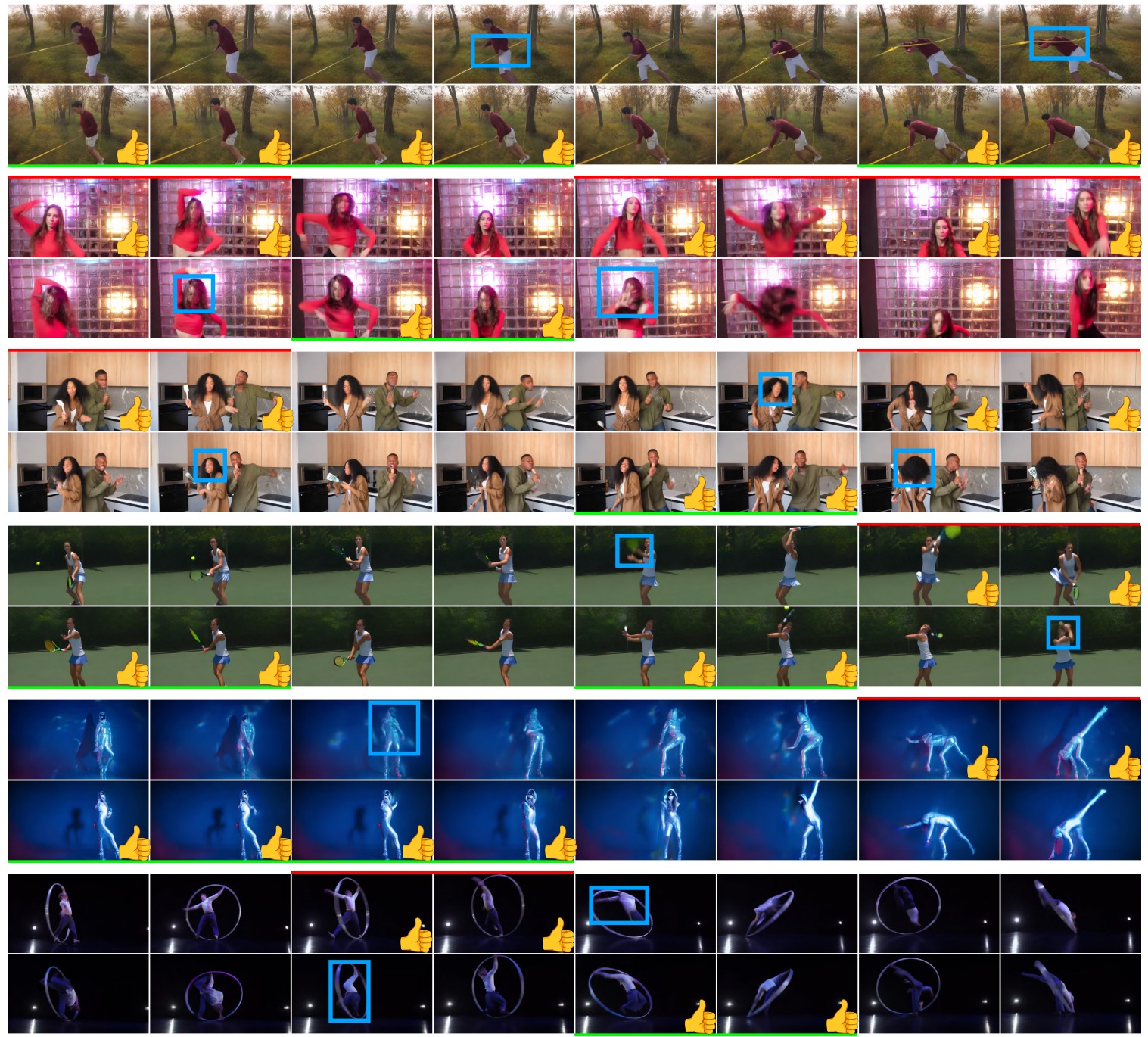

Figure 11: **Uncurated samples of GPT o3 predicted dense preference labels.** Each sample consists of a pair of structurally similar videos generated via guided sampling. Videos are sampled at 2 FPS. A top red bar means GPT prefers the first example, and a bottom green bar means GPT prefers the second example, otherwise it is a tie. We highlight some obvious artifacts with blue rectangles.

# E Pytorch-style Pseudo Code for StructuralDPO and DenseDPO

StructuralDPO applies the same Flow-DPO objective as VanillaDPO, which is adopted from [46]:

```python
def flow_dpo_loss(model, ref_model, x_0, x_1, c, l, beta):
    """
    # model: Flow model that takes text embeddings c and timestep t
    #     as inputs and predicts velocity c
    # ref_model: Pre-trained model that is frozen
    # x_0: The first video in the pair, shape [B, T, C, H, W]
    # x_1: The second video in the pair, shape [B, T, C, H, W]
    # c: Text embedding of the prompt
    # l: Preference label, shape [B], each item is either +1 or -1
    #     +1 means x_0 is better than x_1, -1 means the other way
    # beta: DPO regularization hyper-parameter
    # returns: Flow-DPO loss value
    """
    # Add noise to videos
    t = logit_normal_sampler(x_0.shape[0])
    noise = torch.randn_like(x_0)
    noisy_x_0 = (1 - t) * x_0 + t * noise
    noisy_x_1 = (1 - t) * x_1 + t * noise

    # Compute velocity prediction loss
    v_0_pred = model(noisy_x_0, c, t)
    v_1_pred = model(noisy_x_1, c, t)
    v_ref_0_pred = ref_model(noisy_x_0, c, t)
    v_ref_1_pred = ref_model(noisy_x_1, c, t)
    v_0_gt = noise - x_0
    v_1_gt = noise - x_1
    model_0_err = ((v_0_pred - v_0_gt) ** 2).mean(dim=[1, 2, 3, 4])
    model_1_err = ((v_1_pred - v_1_gt) ** 2).mean(dim=[1, 2, 3, 4])
    ref_0_err = ((v_ref_0_pred - v_0_gt) ** 2).mean(dim=[1, 2, 3, 4])
    ref_1_err = ((v_ref_1_pred - v_1_gt) ** 2).mean(dim=[1, 2, 3, 4])

    # Compute DPO loss
    diff_0 = model_0_err - ref_0_err  # Shape [B]
    diff_1 = model_1_err - ref_1_err  # Shape [B]
    inside_term = -0.5 * beta * l * (diff_0 - diff_1)
    loss = -1 * log(sigmoid(inside_term)).mean()
    return loss
```

DenseDPO extends the Flow-DPO loss to frame-level (or token-level for latent models):

```python
def flow_dense_dpo_loss(model, ref_model, x_0, x_1, c, l, beta):
    """
    # model: Flow model that takes text embeddings c and timestep t
    #     as inputs and predicts velocity c
    # ref_model: Pre-trained model that is frozen
    # x_0: The first video in the pair, shape [B, T, C, H, W]
    # x_1: The second video in the pair, shape [B, T, C, H, W]
    # c: Text embedding of the prompt
    # l: Dense preference label, shape [B, T], can be +1, 0, or -1
    #     +1 means x_0 is better than x_1, -1 means the other way
    #     0 means a tie
    # beta: DPO regularization hyper-parameter
    # returns: Flow-DPO loss value
    """
    # Add noise to videos
    t = logit_normal_sampler(x_0.shape[0])
    noise = torch.randn_1ike(x_0)
    noisy_x_0 = (1 - t) * x_0 + t * noise
    noisy_x_1 = (1 - t) * x_1 + t * noise

    # Compute velocity prediction loss
    v_0_pred = model(noisy_x_0, c, t)
    v_1_pred = model(noisy_x_1, c, t)
    v_ref_0_pred = ref_model(noisy_x_0, c, t)
    v_ref_1_pred = ref_model(noisy_x_1, c, t)
    v_0_gt = noise - x_0
    v_1_gt = noise - x_1
    model_0_err = ((v_0_pred - v_0_gt) ** 2).mean(dim=[2, 3, 4])
    model_1_err = ((v_1_pred - v_1_gt) ** 2).mean(dim=[2, 3, 4])
    ref_0_err = ((v_ref_0_pred - v_0_gt) ** 2).mean(dim=[2, 3, 4])
    ref_1_err = ((v_ref_1_pred - v_1_gt) ** 2).mean(dim=[2, 3, 4])

    # Compute DPO loss
    diff_0 = model_0_err - ref_0_err  # Shape [B, T]
    diff_1 = model_1_err - ref_1_err  # Shape [B, T]
    inside_term = -0.5 * beta * l * (diff_0 - diff_1)
    inside_term = inside_term[l != 0]  # Only take non-tie frames
    loss = -1 * log(sigmoid(inside_term)).mean()
    return loss
```

