# OpenReview forum: "DenseDPO: Fine-Grained Temporal Preference Optimization for Video Diffusion Models"
_NeurIPS.cc/2025/Conference — NeurIPS 2025 spotlight_

### Official Review · Reviewer_PkkZ · 2025-06-01

**Clarity:** 3
**Significance:** 2
**Originality:** 3
**Rating:** 4
**Confidence:** 4

**Summary:**

This paper presents DenseDPO, a framework designed to improve preference optimization in text-to-video diffusion models. Building upon prior work on Direct Preference Optimization (DPO), the authors identify two key limitations in existing video DPO pipelines: (1) a bias toward low-motion video clips due to independent noise sampling, and (2) coarse, whole-video preference labels. DenseDPO addresses these by: (1) Using partially noised real videos to generate structurally similar pairs (StructuralDPO), mitigating motion bias. (2) Introducing segment-level (fine-grained) preference annotations to create richer learning signals. Extensive quantitative evaluations (VideoJAM-bench, MotionBench) and human studies show DenseDPO outperforms VanillaDPO and StructuralDPO in generating high-quality, dynamic, and text-aligned videos.

**Questions:**

See weakness.

**Ethical Concerns:**

["NO or VERY MINOR ethics concerns only"]

**Final Justification:**

The authors' response solved my concern. I tend to increase my score and lean to borderline acceptance.

**Limitations:**

yes.

**Paper Formatting Concerns:**

n/a.

**Quality:**

2

**Strengths And Weaknesses:**

Strength:

1) Clear problem formulation: The paper clearly identifies and investigates the motion bias issue inherent in vanilla DPO setups for post-training video generative models. This is a well-motivated and important problem for post-train the video generative model.
2) The method demonstrates compelling results across multiple challenging benchmarks, complemented by qualitative examples.

Weakness:

1) Reliance on real-video-guided generation: DenseDPO uses partially noised real videos to create structurally similar pairs, which may reduce motion diversity. This raises the question: could this approach limit the ability to select or generate more diverse and high-quality motion?
2) Terminological ambiguity: The introduction of both StructuralDPO and DenseDPO could be confusing. Since DenseDPO builds upon StructuralDPO by adding segment-level labels, the distinction between the two might be better integrated or clarified to reduce redundancy and improve conceptual coherence.
3) Limitations of image-based VLMs: The use of Vision-Language Models (GPT-o3), which are primarily trained on static images, raises concerns about their effectiveness in evaluating motion. It’s unclear whether their judgments reflect true motion quality or simply frame-level visual quality.

---

> ### Author Rebuttal · Authors · 2025-07-30
>
> We thank the reviewer for the constructive feedback. We are glad that the reviewer finds our paper well-motivated and our results compelling across multiple challenging benchmarks. We address the concerns below:
>
> ### **W1. Reliance on real-video-guided generation and its effect on learning high-quality motion.**
>
> A: This is an interesting question. We agree that in theory, VanillaDPO independently samples two videos with distinct motion, which should ideally give more diverse learning signals. However, on one hand, human labeling suffers from a motion bias that favors high-quality yet slow-motion videos. As a result, VanillaDPO often biases the model towards generating static motion, further reducing the motion diversity. On the other hand, our training videos are filtered to contain high-quality motion. It is enough to just learn from the distribution of motion in them.
>
> In DenseDPO, we intentionally choose to hold the motion dynamics while improving visual quality and temporal consistency. We believe the significantly improved motion smoothness of videos generated by our model proves that it learns high-quality motion in training videos.
>
> At a higher level, recently there is still debate about whether post-training teaches models new knowledge, or just amplifies the behaviors learned in pre-training [1, 2]. For DPO methods, we can view it as re-allocating the model’s distribution density to the modes we, as humans, want to see. For example, if before DPO, the model can generate high-quality motion only once in 100 samples, then after DPO, high-quality motion will appear much more frequently. The main goal of DPO is to incentivize high-quality motion presented in training videos. If a type of motion is not in training data and thus not learned by the model in pre-training, it is likely impossible to learn it in post-training either. Therefore, relying on high-quality real videos does not hurt the general DPO objective.
>
> [1] Yue, Yang, et al. "Does reinforcement learning really incentivize reasoning capacity in LLMs beyond the base model?." arXiv 2025.
>
> [2] Zhao, Rosie, et al. "Echo chamber: RL post-training amplifies behaviors learned in pretraining." arXiv 2025.
>
>
> ### **W2. Terminological ambiguity between StructuralDPO and DenseDPO.**
>
> A: Thanks for this suggestion. As the reviewer correctly pointed out, DenseDPO builds upon StructuralDPO by adding segment-level labels, which is only possible with structurally similar videos since they are temporally aligned. We can also rename StructuralDPO to a variant of DenseDPO whose segment length $s$ equals to the video length. We will add more discussions and clarifications to the main text of the paper. Please let us know if you have better suggestions on how to make their connections and distinctions clearer!
>
>
> ### **W3. Limitations of GPT-o3 in evaluating videos.**
>
> A: We kindly note that **we did not use GPT-o3 to measure quantitative results**. Instead, we use VBench [22], a benchmark designed specifically for videos, and VisionReward [61], a video language model fine-tuned to assess video quality, for automatic metrics. They are widely adopted in the literature of video generation and reflect both visual and motion quality of videos. In addition, we also provide user studies that show highly consistent results with the quantitative metrics. This validates the performance gain of DenseDPO.
>
> Regarding GPT-o3’s ability to evaluate videos, we believe our VLM label experiments in Sec.4.3 provide some insights. For ease of discussion, we summarize some key results in the table below. First of all, we agree with the reviewer that since GPT-o3 is primarily trained on static images, it fails to assess long-range motion in 5s videos. As shown in the “Preference Accuracy” column, GPT-o3 only achieves 53.45\% preference prediction accuracy when evaluating 5s videos. Yet, we observe that GPT-o3 works fairly well in assessing short 1s video segments since they only contain modest motion. When tasked with our proposed segment-level preference labeling, GPT-o3 achieves 70.59\% accuracy, even surpassing VisionReward, a video reward model specifically fine-tuned for this task. In addition, we also tried running DPO with VLM predicted preference labels. DenseDPO with GPT-o3 segment-level labels clearly improves several metrics on our video benchmark, and outperforms StructuralDPO with VisionReward. Overall, this proves the effectiveness of our method in adapting off-the-shelf VLMs primarily trained on images to improve video assessment and generation. This is one of the advantages of DenseDPO compared to previous DPO methods.
>
>
> | VLM | Preference Accuracy | Text Alignment | Visual Quality | Temporal Consistency | Dynamic Degree |
> |-|-|-|-|-|-|
> | Pre-trained | - | 0.770 | 0.192 | 0.354 | **0.680** |
> | VisionReward + StructuralDPO | *62.11\%* | *0.785* | *0.347* | *0.521* | 0.622 |
> | GPT-o3 + StructuralDPO | 53.45\% | 0.759 | 0.284 | 0.506 | 0.621 |
> | **GPT-o3 Segment + DenseDPO** | **70.59\%** | **0.842** | **0.368** | **0.598** | *0.672* |

---

> > ### Comment · Reviewer_PkkZ · 2025-08-05
> >
> > The authors' response solved my concern. I tend to increase my score and lean to borderline acceptance.

---

> > > ### Author Response · Authors · 2025-08-05
> > > **Thank you for your consideration!**
> > >
> > > Dear Reviewer PkkZ,
> > >
> > > We thank the reviewer for the positive feedback and we are glad that the concerns have been addressed. We will revise the manuscript accordingly based on the suggestions. We appreciate your thoughtful comments and constructive feedback for the work.
> > >
> > > Best Regards,
> > >
> > > Authors of Submission14593

---

### Official Review · Reviewer_jRmA · 2025-07-02

**Clarity:** 4
**Significance:** 3
**Originality:** 4
**Rating:** 5
**Confidence:** 5

**Summary:**

This work proposes DenseDPO, which first creates aligned video pairs with different local motion details and then labels segment-level preferences to yield a dense learning signal for physics-realistic motion reinforcement learning. It also shows that DenseDPO trained with VLMs for segment-level preferences achieves performance comparable to using human labels. The experiments are thorough and validate the effectiveness of each component.

**Questions:**

See Weaknes

**Ethical Concerns:**

["NO or VERY MINOR ethics concerns only"]

**Final Justification:**

My concerns have been well resolved, so I intend to increase my score to Accept. Also, I think the authors have done enough experiments during the rebuttal period, especially comparisons to other RL methods and scaling the VLM reward. I appreciate this feedback.

**Limitations:**

Yes

**Paper Formatting Concerns:**

no formatting concerns

**Quality:**

3

**Strengths And Weaknesses:**

Strengths

1. This work first proposes StructuralDPO, which generates each pair by denoising partially noised real videos under different guidance strengths. This produces video pairs with shared structural similarity but differing in local details—unlike vanilla DPO’s independent noise samples.

2. By collecting and optimizing with segment-level preference data, DenseDPO achieves clear gains: qualitative examples show markedly improved motion fidelity, and quantitative results on VBench metrics outperform both SFT and vanilla DPO. The appendix provides a mathematical derivation of the guided sampling scheme, explaining why StructuralDPO alone falls short and why DenseDPO prevails.

3. The paper demonstrates that off-the-shelf VLMs (notably GPT) can predict segment-level preferences nearly as well as humans. This removes reliance on costly human labels and lays a solid foundation for scaling post-training temporal dense alignment.

Weaknesses

1. Unclear base model and high post-training cost. The paper omits details on the base diffusion model’s architecture and size. Post-training on 50K videos requires approximately 16 hours on 64 GPUs, imposing substantial computational overhead. Moreover, with code, data, and pretrained checkpoints unreleased, reproducing or evaluating these results is very challenging.

2. Limited novelty in the RL formulation.
Most gains stem from segment-level preference labels rather than any new reinforcement-learning algorithm. The algorithmic contribution over existing DPO variants is therefore modest.

3. Unexplained VLM label noise compensation.
GPT-3.5 labels achieve only ~70 % segment-level accuracy yet match human-label performance in DPO fine-tuning. The manuscript does not analyze why noisy but abundant VLM labels suffice—for example, whether sheer data volume outweighs reduced label quality.

---

> ### Author Rebuttal · Authors · 2025-07-30
>
> We thank the reviewer for the detailed feedback and the positive assessment of our work. We are glad that the reviewer accurately identifies the advantages of StructuralDPO and DenseDPO, as well as highlighting our mathematical derivation and VLM results. We address the concerns below:
>
> ### **W1. Details on models & Training cost & Reproducibility of the work.**
>
> A: We kindly refer the reviewer to Appendix A.2 (the “Pre-trained model” paragraph) for details on the base diffusion model’s architecture. Overall, it is a DiT model similar to Open-Sora [19]. The size of the model is around 10B.
>
> Regarding the training cost, one can easily adapt our method to train on 8 GPUs with gradient accumulation. We also note that all baselines are trained with a similar amount of compute, ensuring a fair comparison.
>
> Regarding code release, we are working on releasing a reference implementation of StructuralDPO and DenseDPO. We also note that we have provided all implementation details in Appendix A such as the GPT prompts to select high-quality data, and the pseudo code in Appendix E, which is only a few lines different from Diffusion-DPO [53] and should be easy to re-implement and try out.
>
>
> ### **W2. Novelty of our algorithm.**
>
> A: We agree that the DPO loss we use is similar to the one proposed by Diffusion-DPO [53]. Yet, this paper focuses on another important component in the DPO framework: how to create better human preference labels. We believe it offers novel insights in designing a better DPO method for video diffusion models, which are not obvious from previous work:
> - We are the first work that explicitly points out the **motion bias** in VanillaDPO, and adapts guided paired video generation to solve this issue in StructuralDPO.
> - As the reviewer correctly pointed out, we provide **mathematical derivation** to show why StructuralDPO is sub-optimal. This aligns with previous observations in LLM DPO papers [40, 47] and serves as a missing piece in diffusion DPO studies.
> - With this insight, we further propose to use **segment-level preference labels** to provide dense learning signals, achieving better generation results and data efficiency compared to baselines. The use of segment-level preference optimization has never been explored in prior diffusion DPO papers.
> - Please note that labeling dense preferences is only possible with structurally similar videos, as it requires the video pairs to be temporally aligned. It also opens up new possibilities to leverage off-the-shelf VLMs for preference learning.
>
> Overall, these design choices and results were neither known nor obvious for video DPO before our work, and we view them as our major contributions driving the performance gains. We believe our work has the potential to impact both the video generation and diffusion post-training communities.
>
>
> ### **W3. VLM label noise compensation.**
>
> A: Thank you for this great question. We conducted additional experiments to compare DenseDPO trained with different amounts of GPT o3 labels on VideoJAM-bench. The results are shown in the table below.
>
> | Method | Aesthetic Quality | Imaging Quality | Subject Consistency | Background Consistency | Motion Smoothness | Dynamic Degree | Text Alignment | Visual Quality | Temporal Consistency | Dynamic Degree |
> |-|-|-|-|-|-|-|-|-|-|-|
> | Pretrained | 54.65 | 55.85 | 88.29 | 91.50 | 92.40 | 84.16 | 0.770 | 0.192 | 0.354 | 0.680 |
> | 10k GPT labels | 55.21 | 56.83 | 88.31 | 91.45 | 92.38 | 84.20 | 0.782 | 0.247 | 0.360 | 0.668 |
> | 35k GPT labels | 55.89 | 58.75 | 89.68 | 92.25 | 92.70 | 84.89 | 0.817 | 0.316 | 0.498 | 0.670 |
> | 55k GPT labels | 56.23 | 60.15 | 90.75 | 93.01 | 92.99 | 85.21 | 0.842 | 0.368 | 0.598 | 0.672 |
>
> Since VLM labels are noisy, DenseDPO with 10k GPT labels achieves only marginal improvement compared to the pre-trained model. Nevertheless, **as the amount of VLM labels grows, we see consistent improvement in all metrics**, closing the gap with human labels. This trend reveals that the quantity of VLM labels indeed compensate for their quality. We view this as a promising future direction to further scale up automatic preference learning with more data and better off-the-shelf VLMs. We will add these results to the paper.

---

> > ### Author Response · Authors · 2025-08-05
> > **Official Comment by Authors**
> >
> > Dear Reviewer jRmA,
> >
> > Thank you again for your time and effort in reviewing our paper. We are really glad to see your initial positive feedback of the paper. We also sincerely appreciate your constructive comments and suggestions, which have helped us improve our work.
> >
> > We would be happy to clarify any points that may help in your evaluation. Please feel free to reach out during the discussion period if you have any questions or thoughts.
> >
> > Best regards,
> >
> > Authors of Paper #14593

---

> > ### Comment · Reviewer_jRmA · 2025-08-06
> >
> > My concerns have been well resolved, so I intend to increase my score to Accept. Also, I think the authors have done enough experiments during the rebuttal period, especially comparisons to other RL methods and scaling the VLM reward. I appreciate this feedback.

---

> > > ### Author Response · Authors · 2025-08-07
> > > **Thank you!**
> > >
> > > Dear Reviewer jRmA,
> > >
> > > We are glad to know that the concerns have been addressed and we thank the reviewer for raising the score. We will revise the manuscript accordingly based on the suggestions. We appreciate your thoughtful comments and constructive feedback for the work.
> > >
> > > Best Regards,
> > >
> > > Authors of Submission14593

---

### Official Review · Reviewer_BRHs · 2025-07-03

**Clarity:** 3
**Significance:** 2
**Originality:** 2
**Rating:** 4
**Confidence:** 3

**Summary:**

The paper proposes DenseDPO, which provides fine-grained segment-level annotations for DPO.
Ground-truth videos are used to construct video pairs for training.
The motion generation results are improved, while text alignment, visual quality, and temporal consistency are maintained.
The authors leverage a VLM to automatically score the video pairs and show that the results are comparable to human annotations.

**Questions:**

see weakness

**Ethical Concerns:**

["NO or VERY MINOR ethics concerns only"]

**Final Justification:**

I have gone through the author's rebuttal and other reviewers' comments. The rebuttal and explainations have almost address my concerns. However, I still have concerns on the method's novelty. Considering other reviewers' comments I decide to raise my score.

**Limitations:**

yes

**Paper Formatting Concerns:**

No obvious formatting issues

**Quality:**

3

**Strengths And Weaknesses:**

Strengths
The writing is generally good, and the experiments are comprehensive.
The visual results demonstrate large motion.

Weaknesses
The proposed method, DenseDPO, has limited technical novelty. Each component offers only minor innovation.
The results are not strong: in some cases, the performance is worse than vanilla DPO, and the degree of motion dynamics is lower than that of the pretrained baseline.
There is a lack of ablation studies on the use of ground-truth data.
The results on motion smoothness and motion dynamics in MotionBench are worse than those of vanilla DPO and SFT, making it difficult to support the paper’s claim of improved motion generation.

---

> ### Author Rebuttal · Authors · 2025-07-30
>
> We are glad that the reviewer found our writing generally good, our experiments comprehensive, and the visual results showing large motion. We address the concerns below:
>
> ### **W1 & W3. Results on motion smoothness and motion dynamics compared to baselines.**
>
> A: We would like to point out that motion smoothness or motion dynamics alone does not represent the video quality. For example, an entirely static video will have perfect motion smoothness, while a video of completely random noise will have high motion dynamics. For video generators, there is an inherent trade-off between the two [35, 61]. **Please check out the video results in our supplementary material for a clear comparison**. Specifically,
> - VanillaDPO achieves high motion smoothness because it generates **almost static videos**. This can be verified by its low motion dynamics score. In addition, DenseDPO actually achieves a higher motion smoothness than VanillaDPO on VideoJAM-bench, and only slightly underperforms on MotionBench.
> - Pre-trained and SFT models achieve high motion dynamics but generate videos with **severe temporal inconsistency**, such as deformed limbs and objects. This can be verified by their low motion smoothness score. Besides, DenseDPO also outperforms baselines on VideoJAM-bench, and the gap of motion dynamics is marginal on MotionBench.
> - Overall, DenseDPO is the only method that maintains the motion dynamics of the pre-trained model, while significantly improving the motion smoothness. We would also like to note that other reviewers find our results “strong” while using “fewer labeled videos” (RyCj), “validate the effectiveness of each component” (jRmA), and “compelling across multiple challenging benchmarks” (PkkZ).
>
> We also note that all automatic metrics in video generation have limitations [1, 2], and **human evaluations are considered more reliable**:
> - Fig.5 right of the main paper shows that DenseDPO achieves slightly higher motion smoothness (TC) with VanillaDPO, with significantly better motion dynamics (DD).
> - Fig.3 right of the Appendix shows that DenseDPO achieves slightly better motion dynamics (DD) as the pre-trained model, while improving motion smoothness (TC) by a sizable margin.
>
> [1] ByteDance. "Seaweed-7B: Cost-effective training of video generation foundation model." arXiv 2025.
>
> [2] Zeng, Ailing, et al. "The dawn of video generation: Preliminary explorations with sora-like models." arXiv 2024.
>
>
> ### **W1. Novelty of DenseDPO.**
>
> A: We agree that both DPO and guided generation were known individually in the diffusion literature. However, this paper offers novel insights in **designing a better DPO method for video diffusion models**, which are not obvious from previous work:
> - We are the first work that explicitly points out the **motion bias** in VanillaDPO, and adapts guided generation to solve this issue with StructuralDPO.
> - Instead of naively borrowing the technique, we provide **mathematical derivation** to show why StructuralDPO is sub-optimal (see Appendix B). This aligns with previous observations in LLM DPO papers [40, 47] and serves as a missing piece in diffusion DPO study.
> - With this insight, we further propose to use **segment-level preference labels** to provide dense learning signals, achieving better generation results and data efficiency compared to baselines. The use of dense preferences is never explored in prior video DPO papers.
> - Besides the DenseDPO framework, we also want to highlight our results using **VLM labels**. It shows a promising way to scale up automatic preference learning with off-the-shelf VLMs.
>
> Overall, these design choices and results were neither known nor obvious for video DPO before our work, and we view them as our major contributions driving the performance gains. We believe our work has the potential to impact both the video generation and diffusion post-training communities.
>
>
> ### **W2. Ablation on the use of ground-truth data.**
>
> A: We are not entirely sure what the reviewer means by “use of ground-truth data”. Based on context, we assume the reviewer is asking about ablating the ground-truth videos used in guided video generation.
>
> In DenseDPO, we randomly sample 10k videos from our 55k high-quality dataset to serve as guidance videos. Here, we randomly sample another 10k videos without overlap with the previously selected ones, use them to generate paired videos, and conduct a DenseDPO training. We report the result on VideoJAM-bench below. Both runs achieve similar results, proving that our method is robust to the selection of ground-truth data as long as they are high-quality videos.
>
> | Method | Aesthetic Quality | Imaging Quality | Subject Consistency | Background Consistency | Motion Smoothness | Dynamic Degree | Text Alignment | Visual Quality | Temporal Consistency | Dynamic Degree |
> |-|-|-|-|-|-|-|-|-|-|-|
> | DenseDPO   (old GT data) | 56.99 | 60.92 | 91.54 | 93.84 | 93.56 | 85.38 | 0.863 | 0.376 | 0.632 | 0.680 |
> | DenseDPO   (new GT data) | 56.94 | 60.91 | 91.48 | 93.76 | 93.57 | 85.39 | 0.868 | 0.377 | 0.636 | 0.678 |
>
> Please let us know if this is the correct interpretation. We are happy to provide extra ablations or results if needed. We will add more discussions and the results above to the paper.

---

> > ### Author Response · Authors · 2025-08-05
> > **Official Comment by Authors**
> >
> > Dear Reviewer BRHs,
> >
> > Thank you again for your time and effort in reviewing our paper. We sincerely appreciate your constructive comments and suggestions, which have helped us improve our work.
> >
> > We would be happy to clarify any points that may help in your evaluation. Please feel free to reach out during the discussion period if you have any questions or thoughts.
> >
> > Best regards,
> >
> > Authors of Paper #14593

---

> ### Author Response · Authors · 2025-08-08
>
> Dear Reviewer BRHs,
>
> Thank you again for your time and effort in reviewing our paper. We sincerely appreciate your constructive comments and suggestions, which have helped us improve our work. We have addressed all your concerns in the rebuttal, including clarifications on quantitative results, the novelty of DenseDPO, and additional ablation studies.
>
> Since the discussion period is coming to an end, we would like to double-check if you have any remaining questions about our work. Please feel free to reach out if you have any thoughts. We are happy to provide more discussions.
>
> Best regards,
>
> Authors of Paper #14593

---

> > ### Comment · Reviewer_BRHs · 2025-08-08
> >
> > I have gone through the author's rebuttal and other reviewers' comments.
> > The rebuttal and explainations have almost address my concerns. However, I still have concerns on the method's novelty. Considering other reviewers' comments I decide to raise my score.

---

> > > ### Author Response · Authors · 2025-08-08
> > > **Thank you for your considerations!**
> > >
> > > We thank the reviewer for their thoughtful consideration of our rebuttal and for raising their score. We are glad that our explanations have addressed most of the concerns.
> > >
> > > We would like to briefly re-emphasize that our core contribution lies in designing a DPO algorithm tailored for video diffusion models. To this end, we analyzed the motion bias in existing methods, introduced dense preferences, and scaled up VLM-based labels, all of which were previously underexplored in this context.
> > >
> > > We will ensure these points are presented more clearly in the revised manuscript. We sincerely appreciate the reviewer’s time, constructive feedback, and support.
> > >
> > > Best Regards,
> > >
> > > Authors of Submission14593

---

### Official Review · Reviewer_RyCj · 2025-07-11

**Clarity:** 3
**Significance:** 2
**Originality:** 3
**Rating:** 4
**Confidence:** 4

**Summary:**

The paper proposes DenseDPO, a method to improve video diffusion models by using segment-level preference labels instead of whole-video comparisons. It generates structurally similar video pairs and collects fine-grained human or VLM-based preferences, avoiding bias toward slow-motion clips. DenseDPO achieves better motion quality, matches other methods in visual fidelity and text alignment, and is more data-efficient. Therefore, it needs fewer labeled videos to perform well.

**Questions:**

1. Can you provide comparisons with more methods? The experiments can be done using a small data scale.
2. Can you provide the results that directly utilize the VLM rewards to train the generation model?

**Ethical Concerns:**

["NO or VERY MINOR ethics concerns only"]

**Final Justification:**

The explanation solve the problem I mentioned in the weaknesses.

**Limitations:**

1. The upper bound of the method is limited by the data and the quality of annotations, which means the method can’t be easily scaled.
2. Lack of comparisons with many methods

**Quality:**

3

**Strengths And Weaknesses:**

Strengths:

1. Segment-level preferences capture more accurate feedback than global labels.
2. Using structurally aligned video pairs avoids overfitting to static, artifact-free clips.
3. It achieves strong results with only one-third of the labeled data.

Weaknesses：

1. The method is more likely to find different reward aspects for DPO. If so, why not directly use the reward model for online RL?
2. Missing comparisons with many DPO-like methods, such as D3PO, SePPO, DPOK, etc.

---

> ### Author Rebuttal · Authors · 2025-07-30
>
> We thank the reviewer for acknowledging the benefit of using structurally similar videos, segment-level preference labels, and the data-efficiency of DenseDPO. We address the concerns below:
>
> ### **W1. Directly use the reward model for online RL.**
>
> A: We kindly remind the reviewer that the major goal of this work is to improve the original DPO framework, which relies on pre-collected human preference pairs [46, 53, 62]. Our method demonstrates great performance gain by improving the offline human labeling process. Moreover, in the next question, we will show that DenseDPO outperforms DPOK [15], an online RL method.
>
> Compared to online RL training, DPO bypasses the need for a reward model thus is simple to implement. It also achieves stable improvements and has been applied in several industry lab developed video diffusion models [1, 2]. In addition, existing video reward models are only trained in pixel space. This requires decoding the latent to videos to compute reward, which is both computation and memory intensive. Overall, while RL holds strong potential, the topic of RL for diffusion has only recently been explored [3, 4] and is concurrent to our work.
>
> We also want to point out that effective online RL training relies on good reward models. Yet, even SOTA video reward models are bad at assessing all aspects of a generated video (please see more discussions in the response to Q2). We believe applying online RL for video diffusion models requires developing better reward models [4], which is beyond the scope of this work.
>
> [1] Chen, Guibin, et al. "SkyReels-V2: Infinite-length film generative model." arXiv 2025.
>
> [2] Ma, Guoqing, et al. "Step-Video-T2V Technical Report: The practice, challenges, and future of video foundation model." arXiv 2025.
>
> [3] Liu, Jie, et al. "Flow-GRPO: Training flow matching models via online rl." arXiv 2025.
>
> [4] Xue, Zeyue, et al. "DanceGRPO: Unleashing GRPO on Visual Generation." arXiv 2025.
>
>
> ### **W2 & Q1 & L2. Comparison with more baselines such as D3PO, SePPO, DPOK.**
>
> A: Thank you for bringing up these methods. To provide comparisons, we adapted D3PO’s public code to our video setting. For DPOK, we also modified their open-source code and leveraged VisionReward [61] to provide online feedback. Since no training code is available for SePPO, we re-implement it based on its paper. All methods are trained on the same data as VanillaDPO (the human preference pairs and text prompts) with LoRA and the hyper-parameters listed in their original papers.
>
> We report the results on VideoJAM-bench in the table and summarize them below:
> - D3PO achieves similar results as VanillaDPO which is based on Diffusion-DPO. This is in line with prior works [1, 2] that also report similar performances between the two methods.
> - DPOK, an online RL method, slightly improves the visual quality and temporal consistency compared to the pre-trained model, yet significantly degrades the dynamic degree. This is because VisionReward is biased towards per-frame visual quality instead of temporal motion. Thus, tuning with its noisy reward leads to static motion. Please see more discussions about the reward model in the next question.
> - SePPO [1] achieves a better dynamic degree than VanillaDPO, yet performs worse in visual quality. Instead of using offline generated losing samples, SePPO generates online samples with a reference model, and designs a filtering method (AAF) to assess its quality. This can be viewed as a combination of DPO and SFT. However, we noticed that AFF is based on model denoising loss, which is sometimes unreliable. In those cases, the model is optimized on low-quality videos, leading to worse visual quality.
>
> Overall, DenseDPO outperforms D3PO, DPOK, and SePPO across all metrics, showing the effectiveness of our approach. We will add these results and discussions to the paper.
>
> | Method | Aesthetic Quality | Imaging Quality | Subject Consistency | Background Consistency | Motion Smoothness | Dynamic Degree | Text Alignment | Visual Quality | Temporal Consistency | Dynamic Degree |
> |-|-|-|-|-|-|-|-|-|-|-|
> | Pretrained | 54.65 | 55.85 | 88.29 | 91.50 | 92.40 | 84.16 | 0.770 | 0.192 | 0.354 | **0.680** |
> | D3PO | 56.15 | 58.03 | 88.93 | 92.23 | 92.78 | 82.53 | 0.833 | 0.322 | 0.482 | 0.602 |
> | DPOK | 54.99 | 56.28 | 89.14 | 92.36 | 92.95 | 78.65 | 0.795 | 0.337 | 0.518 | 0.457 |
> | SePPO | 55.83 | 57.42 | 89.93 | 92.65 | 92.93 | *82.85* | 0.841 | 0.326 | 0.554 | 0.587 |
> | VanillaDPO | **57.25** | *60.38* | *91.21* | **93.94** | *93.43* | 80.25 | **0.867** | *0.371* | **0.636** | 0.535 |
> |**DenseDPO**| *56.99* | **60.92** | **91.54** | *93.84* | **93.56** | **85.38** | *0.863* | **0.376** | *0.632* | *0.680* |
>
> [1] Zhang, Daoan, et al. "SePPO: Semi-policy preference optimization for diffusion alignment." arXiv 2024.
>
> [2] Liang, Zhanhao, et al. "Aesthetic post-training diffusion models from generic preferences with step-by-step preference optimization." CVPR 2025.
>
>
> ### **Q2. Results on directly utilizing VLM rewards to train the generation model.**
>
> A: VLM rewards have been used in the literature to improve generation models in different ways: 1) online RL training with VLM rewards, 2) DPO training with preferences derived from VLM rewards, and 3) REFL-style [60] direct gradient backpropagation to maximize the VLM reward. Below, we discuss each direction:
>
> **Online RL training with VLM rewards**: as discussed above, we implemented an online RL algorithm DPOK [15] based on VisionReward [61], and DenseDPO outperforms this baseline.
>
> **DPO with VLM rewards**: we refer the reviewer to Sec.4.3 of the main paper and Appendix C.2 for detailed results. We test SOTA video rewards models fine-tuned from VLMs such as VideoReward [34] and VisionReward [61], as well as advanced commercial VLMs such as GPT o3. We summarize some key findings below:
> - Tab.3 (a) of the main paper shows that even SOTA VLMs achieve low accuracy in predicting human preferences on 5s videos. In contrast, predicting our proposed segment-level preferences leads to higher accuracy.
> - Appendix Tab.1 reveals that VLM-based reward models might be biased towards video content rather than temporal motion, which may explain their low preference accuracy.
> - Tab.3 (b) of the main paper shows that StructuralDPO with VLM rewards achieves moderate improvements compared to the pre-trained model. Applying segment-level VLM rewards further improves the results, which only slightly underperforms DenseDPO with human labels.
>
> **Direct VLM reward maximization**: we tried REFL in our preliminary experiments with HPSv2 [57] and VisionReward [61] as the target reward model. However, we observed severe reward hacking, e.g., the video frames become oversaturated and the temporal motion gets static, which is clearly worse than applying DPO. It also requires huge GPU memory and computation as we need to decode latent tokens to raw pixels to apply these VLMs and backpropagate gradients through them. We will include these results in the paper.
>
>
> ### **L1. The scalability of the proposed method.**
>
> A: While DPO requires pre-collected human preferences, training a reward model for online RL also requires human labels (e.g., VideoReward uses 182k video annotations, and VisionReward uses around 90k video labels). In fact, a post-training stage with high-quality human-labeled data has been a common practice in both diffusion models [1, 2] and LLMs [3, 4]. In addition, we kindly note that DenseDPO is already more data-efficient than DPO baselines.
>
> In the paper, we also explored VLM labels for scalable DPO training. We conducted additional experiments comparing DenseDPO trained with different amounts of GPT o3 labels on VideoJAM-bench and show the results below.
>
> | Method | Aesthetic Quality | Imaging Quality | Subject Consistency | Background Consistency | Motion Smoothness | Dynamic Degree | Text Alignment | Visual Quality | Temporal Consistency | Dynamic Degree |
> |-|-|-|-|-|-|-|-|-|-|-|
> | Pretrained | 54.65 | 55.85 | 88.29 | 91.50 | 92.40 | 84.16 | 0.770 | 0.192 | 0.354 | 0.680 |
> | 10k GPT labels | 55.21 | 56.83 | 88.31 | 91.45 | 92.38 | 84.20 | 0.782 | 0.247 | 0.360 | 0.668 |
> | 35k GPT labels | 55.89 | 58.75 | 89.68 | 92.25 | 92.70 | 84.89 | 0.817 | 0.316 | 0.498 | 0.670 |
> | 55k GPT labels | 56.23 | 60.15 | 90.75 | 93.01 | 92.99 | 85.21 | 0.842 | 0.368 | 0.598 | 0.672 |
>
> As the amount of VLM labels grows, we see consistent improvement in all metrics. This trend clearly shows that our method scales well with the quantity of VLM labels. Moreover, VLM itself is an actively evolving field, and our method will benefit from its progress. We would also like to note that reviewer jRmA recognized these results as “removes reliance on costly human labels and lays a solid foundation for scaling post-training temporal dense alignment”. We will add the discussion on the data and the quality of annotations to the Limitation section of the paper.
>
> [1] ByteDance. "Seaweed-7B: Cost-effective training of video generation foundation model." arXiv 2025.
>
> [2] StepFun. "Step-Video-T2V Technical Report: The practice, challenges, and future of video foundation model." arXiv 2025.
>
> [3] DeepSeek. "DeepSeek-R1: Incentivizing reasoning capability in LLMs via reinforcement learning." arXiv 2025.
>
> [4] Meta. "The Llama 4 herd". Meta 2025.

---

> > ### Author Response · Authors · 2025-08-05
> > **Official Comment by Authors**
> >
> > Dear Reviewer RyCj,
> >
> > Thank you again for your time and effort in reviewing our paper. We sincerely appreciate your constructive comments and suggestions, which have helped us improve our work.
> >
> > We would be happy to clarify any points that may help in your evaluation. Please feel free to reach out during the discussion period if you have any questions or thoughts.
> >
> > Best regards,
> >
> > Authors of Paper #14593

---

> > ### Comment · Reviewer_RyCj · 2025-08-08
> > **Feedback**
> >
> > Thanks for the explanation, I will raise my score.

---

> > > ### Author Response · Authors · 2025-08-08
> > > **Thank you!**
> > >
> > > Dear Reviewer RyCj,
> > >
> > > Thank you so much for raising your score! We will incorporate all additional experiments and discussions into the paper. We appreciate your thoughtful comments and constructive feedback for the work.
> > >
> > > Best Regards,
> > >
> > > Authors of Submission14593

---

> ### Comment · Reviewer_jRmA · 2025-08-06
>
> Dear Reviewer RyCj,
>
> I noticed that the authors have conducted a detailed comparison against several DPO-like methods and have provided VLM labels for scalable DPO training. However, I am not sure whether these experiments address your concerns, especially regarding Weakness #1. Could you please provide feedback to help me determine whether Weakness #1 has now been adequately addressed?
>
> Best regards,
>
> Reviewer jRmA

---

> ### Author Response · Authors · 2025-08-08
>
> Dear Reviewer RyCj,
>
> Thank you again for your time and effort in reviewing our paper. We sincerely appreciate your constructive comments and suggestions, which have helped us improve our work. We have addressed all your concerns in the rebuttal, including comparison with additional baselines such as online RL methods, results on VLM labels, and discussions on the scalability of DenseDPO.
>
> Since the discussion period is coming to an end, we would like to double-check if you have any remaining questions about our work. Please feel free to reach out if you have any thoughts. We are happy to provide more discussions.
>
> Best regards,
>
> Authors of Paper #14593

---

### Author Response · Authors · 2025-08-03
**Official Comment by Authors**

Dear reviewers,

Thank you again for providing insightful feedback to help us improve our work. We have addressed all the questions in the review. Since the discussion period is coming to an end, please kindly let us know if there is anything that needs further clarification. Thank you!

For your convenience, we summarize the additional experiments we conducted during the rebuttal period:
1. In the response to Reviewer RyCj, we benchmark DenseDPO with additional baselines, including D3PO, a DPO-based method, DPOK, an online RL-based method, and SePPO, a hybrid method of both DPO and SFT. Results show that **DenseDPO consistently outperforms all baselines**.
2. In the response to Reviewer RyCj and Reviewer jRmA, we ablate DenseDPO performance when using different amounts of GPT-o3 dense preference labels. Results show that all metrics improve consistently when the number of labels increases, even outperforming task-specifically fine-tuned video reward models. This shows that **our approach can benefit from scaling up VLM label size** and more advanced general-purpose VLMs.
3. In the response to Reviewer BRHs, we ablate the ground-truth videos used in guided video generation. We show that using a different set of videos leads to similar DenseDPO performance. This proves that **the superior performance of DenseDPO is due to our designed labeling process, instead of the selected videos**.

We also provide clarifications to several questions regarding model performance, online RL vs DPO, and the novelty of our work. Please kindly take a look and let us know if there are further questions. Thank you!

Best Regards,

Authors of Submission14593

---

### Decision · Program_Chairs · 2025-09-17

**Decision:**

Accept (spotlight)

**Comment:**

This draft introduces DenseDPO, which uses aligned video pairs and segment-level preferences to improve video diffusion models. The method reduces motion bias, yields stronger motion quality, and remains data-efficient, with promising results even using VLM-based labels. While questions on novelty and reliance on guided videos were raised, AC and Rs agree that the rebuttal provided solid comparisons and clarifications. Overall, the work is well-motivated, convincingly validated, and has a clear impact. AC recommend accept.